# Application and Development of Silicon Anode Binders for Lithium-Ion Batteries

**DOI:** 10.3390/ma16124266

**Published:** 2023-06-08

**Authors:** Huilin Shen, Qilin Wang, Zheng Chen, Changru Rong, Danming Chao

**Affiliations:** 1Key Laboratory of High-Performance Plastics, Ministry of Education, National & Local Joint Engineering Laboratory for Synthesis Technology of High-Performance Polymers, College of Chemistry, Jilin University, Xiuzheng Road 1788, Changchun 130012, China; 2National Key Laboratory of Advanced Vehicle Integration and Control, China FAW Group Co., Ltd., Changchun 130013, China

**Keywords:** lithium-ion batteries (LIBs), silicon anode, polymer binder

## Abstract

The use of silicon (Si) as a lithium-ion battery’s (LIBs) anode active material has been a popular subject of research, due to its high theoretical specific capacity (4200 mAh g^−1^). However, the volume of Si undergoes a huge expansion (300%) during the charging and discharging process of the battery, resulting in the destruction of the anode’s structure and the rapid decay of the battery’s energy density, which limits the practical application of Si as the anode active material. Lithium-ion batteries’ capacity, lifespan, and safety can be increased through the efficient mitigation of Si volume expansion and the maintenance of the stability of the electrode’s structure with the employment of polymer binders. The main degradation mechanism of Si-based anodes and the methods that have been reported to effectively solve the Si volume expansion problem firstly are introduced. Then, the review demonstrates the representative research work on the design and development of new Si-based anode binders to improve the cycling stability of Si-based anode structure from the perspective of binders, and finally concludes by summarizing and outlining the progress of this research direction.

## 1. Background

Lithium-ion batteries (LIBs) are widely used in power supplies and energy storage devices due to their high energy density, long service life, low self-discharge rate and lack of “memory effect” [1,2,3]. With the increasing demand for battery energy storage, the exploration of potential high-specific-capacity anode and cathode materials has become a research hotspot. Early lithium-ion batteries’ cathode materials were transition metal oxides, the most representative being lithium cobaltate (LiCoO_2_). The ternary material Li(Ni,Co,Mn)O_2_, which has a synergistic effect and high energy density with greater electrochemical performance than a single material, is the focus of current research on cathode materials. Graphite has been the primary anode-active material since the commercial release of Sony’s LIBs in 1991, with a theoretical specific capacity of 372 mAh g^−1^ [4]. However, graphite-embedded lithium’s potential is very close to the lithium-ion deposition potential. When the battery is charging, lithium-ions embedded in the graphite anode may deposit on its surface to form lithium dendrites, short-circuiting the battery and posing a serious safety risk [5]. Therefore, to increase the battery’s total energy density, anode materials with high energy density and moderate lithium potential must be found. Silicon (Si) has a high theoretical specific capacity (4200 mAh g^−1^), making it one of the most desirable anode-active materials for LIBs.

## 2. Positive and Negative Characteristics of a Si-Based Anode

To further enhance the energy density of LIBs, researchers have investigated the operating voltages and theoretical specific capacities of a variety of different anode materials, among which Si, as an anode material, has a high theoretical specific capacity and moderate embedded lithium potential. Theoretically, the Si atom can generate Li_22_Si_5_, which has a very high specific capacity (4200 mAh g^−1^), by alloying with 4.4 lithium-ions. Under practical conditions, the Si atom can be alloyed with 3.75 lithium-ions to produce Li_15_Si_4_, with a high practical specific capacity of 3579 mAh g^−1^, which is approximately ten-times that of the graphite anode [6]. In contrast to the graphite anode, a Si-based anode has a somewhat higher embedded lithium potential (0.4 V vs. Li/Li^+^), preventing lithium deposition issues during battery charging and enhancing battery safety [7]. Furthermore, Si is the second-largest element, and relatively common in the universe. Based on the aforementioned benefits, Si has enormous development potential as the active material for lithium-ion batteries [8].

However, when Si is used as the active material in the anode of lithium-ion batteries, its volume expands and shrinks tremendously during the Li^+^ insertion and de-insertion. The specific change process during the volume expansion and contraction of silicon is as follows. When the battery is charged (lithiation process), lithium-ions are embedded in the Si-based anode and undergo an alloying reaction, causing the Si to expand by 300% in volume; when the battery is discharged (de-lithiation process), the lithium-ions are again removed from the Si anode, causing the silicon to contract in volume. As the battery is repeatedly used, the initially perfect electrode structure suffers severe degradation, which results in a considerable decline in performance, or even a spontaneous combustion or explosion, rendering the battery worthless and posing a serious safety risk associated with the battery.

The mechanism of a Si-based anode failure due to Si volume expansion [9] is shown in Figure 1a,b. (1) Firstly, the Si particles experience a remarkable volume expansion during the period of the lithiation process, leading to some degree of silicon particle rupture. The silicon particles then experience a significant volume contraction during the period of the de-lithiation process. After numerous cycles, the Si particles in their initial state will be crushed into smaller particles (Figure 1a) which can easily escape from the electrode and diffuse into the electrolyte, eventually destabilizing the electrode’s structure (Figure 1b). (2) Next, the solid-electrolyte-interface (SEI) layer is formed on the surface of the silicon anode following the initial charge/discharge cycle, a phenomenon which has been shown to have the effect of protecting the electrode from electrolyte corrosion. Nevertheless, the SEI layer on the Si particles breaks down throughout the repetitive charging and discharging process as a result of the Si particles’ substantial expansion and contraction, exposing more fresh surfaces. The increased electrode resistance caused by the thickening SEI layer lowers the battery’s overall performance. In addition, excessive SEI layer-development can consume a lot of lithium-ions in the electrolyte, lowering the cell’s coulombic efficiency [10]. (The aforementioned SEI layer is a passivated layer that has been applied to the electrode’s surface, as a result of numerous complex reactions involving lithium salts, additives, trace air pollutants, etc.; the layer allows the passage of Li^+^ and has the properties of an electrolyte. Its main inorganic and organic components are Li_2_CO_3_, LiF, Li_2_O, LiOH, ROCO_2_Li, (ROCO_2_Li)_2_, ROLi, etc. [11]).

In microscopic state, after repeated cycles of the silicon anode, as shown in Figure 2, the active-substance silicon gradually changes from having a smooth surface to having a dendritic surface. There are large protrusions and cracks on the electrode’s surface which eventually lead to the disintegration of the electrode’s structure.

## 3. Suggestions for Enhancing Si Anode Performance

The following three areas (included in Table 1) have been the focus of work to date to address the volume expansion of Si and maintain the stability of the Si anode structure [10,14,15,16,17,18]. In order to avoid the issue of active material pulverization, Si-based materials should be modified and compounded. Reducing the size of Si particles can lessen the mechanical stress brought on by volume expansion [19]. Additionally, using Si/C composite active materials and silicon oxides (SiO_x_) active materials can lessen the electrode effect on volume expansion during battery charging. The latest development of an active–inactive alloy system is believed to be one of the most effective solutions for boosting silicon anode cycling capacity [12,20]. As well, the addition of carbon material can increase the conductivity of the Si-based electrode [21]. Second, work has centered on the choice of electrolyte additives that can preserve the battery’s capacity density and lengthen battery life by causing the electrode surface to produce the stable and low-resistance SEI layer. Last, but not least, the employment of Si-based anode binders decreases Si volume expansion, reduces active particle and conductive agent shedding, and preserves the stability of the electrode structure. The three concepts mentioned above are further explained below.

Researchers have modulated the size, structure and morphology of silicon-based materials to reduce the size of silicon particles to the nanoscale, effectively reducing the crushing of silicon during the alloying and de-alloying process. However, Si nanomaterials have some serious shortcomings. First of all, the specific capacity of the electrode decreases along with the tap density of Si nano materials. Second, the components made of nano Si frequently agglomerate and disperse ineffectively. Finally, creating materials that are nanoscale in size is challenging and expensive. According to research, using Si/C composites (Figure 3) and SiO_x_ materials can lessen the electrode’s volume expansion [23,28,29]. Si/C composites can also avoid making electrolyte and Si directly come into contact and lessen the electrolyte decomposition brought on by hanging bonds on the silicon’s surface. In recent years, researchers have presented an active–inactive alloy system. The system consists of distributed alloy particles of active amorphous silicon (a-Si) and crystalline iron disilicide (c-FeSi_2_) in graphite. This distinctive multilayer architecture has excellent mechanical properties and a long-term structural stability which can effectively maintain the electrode’s cycling stability during the battery charging-and-discharging processes [12]. The initial cross-section of the electrode is shown in Figure 3c. There are pores inside the electrode that effectively accommodate the expansion of the silicon’s volume. In addition, active material silicon is dispersed uniformly to create a network of connections.

The volume expansion and contraction of Si particles during lithium-ion batteries’ charging and discharging causes the SEI layer between the Si anode and the electrolyte to rupture, exposing new surfaces and re-forming the SEI layer upon electrolyte contact. This process continuously consumes lithium-ions in the electrolyte, and as the SEI layer thickens, the Li^+^ transport to the active material increases the resistance. Moreover, the SEI layer is electron-insulating, and as the SEI layer grows, it will encircle the active material, losing contact with the conductive agent and the current collector. This will result in high resistance and a reversible-capacity fading of the electrode, finally leading to cell failure [15]. Carbonate organic solvents, such as ethylene carbonate (EC), diethyl carbonate (DEC), etc. are the most often-utilized electrolyte solvents [40]. FEC is one of the most studied electrolyte additives for silicon anode-based lithium-ion batteries [41]. Researchers added electrolyte additive FEC to EC and DEC solvents. FEC can react preferentially to form an SEI layer more efficiently than other electrolytes, due to its higher reduction-potential of decomposition. The stable SEI layer limits the appearance of large cracks on the silicon’s surface, and the dense SEI layer minimizes the continuous reaction between anode and electrolyte [17,18]. Therefore, one of the crucial strategies to increase the cycle stability of the Si anode is to use electrolyte additives.

The electrodes of the battery are mainly composed of active materials, conductive additives, polymer binders, etc., as shown in Figure 4 [42]. The active material is used to store lithium-ions, which is a key factor of battery capacity. The conductive additives in the electrode enhance the conductivity of the electrode. The main function of the binder is to firmly bond the active substance, the conductive agent, and the current collector to maintain the stability of the electrode structure in the process of charge and discharge. The binder content is little and devoid of capacity, yet it is crucial to the battery energy density and cycle life [43]. Although binders are employed in greater quantities of 5% to 50% of the electrode material in laboratory settings [14], they are applied in commercial cells at levels less than 5% [44,45]. According to studies, the proper use of a polymer binder may also successfully address the Si anode volume expansion issue, preserve the integrity of the Si anode structure, and produce Si-based LIBs with a high energy density and strong cycling stability.

## 4. Si-Based Anode Polymer Binders

Since 2011, the literature on binders for the Si anode has been increasing year by year [46]. The core function of the polymer binder is to adhere the active particles and conductive agents to the current collector to make the electrode complete. Moreover, the binder for the Si anode also has superior adhesive strength and mechanical properties, as well as other stable properties, minimizing the phenomenon of active particles breaking off from the electrode structure and preserving the stability of the Si anode structure during the use process [47]. The silicon anode polymer binder summarized in Figure 5 is introduced in the review.

### 4.1. Bonding Mechanism of the Binders

The bonding process between the electrode material and the binder is primarily based on mechanical interlocking and interfacial forces [48], and it is divided into two processes called “the de-solution/diffusion/penetration” step and the “hardening” step. As shown in Figure 6a, the “penetration process” is performed with the help of the solvent in the wet electrode process. The polymer binder first infiltrates the surface of the electrode materials and penetrates through the micropores on the surface of the electrode materials into the complex tortuous voids inside the active material. During the “penetration process”, the binder is in full contact with the electrode material. As shown in Figure 6b, in the wet electrode process, as the evaporation of the solvent (“hardening process”), the mechanical interlocking effect occurs between the polymer binder and the surface of the electrode material. There are two types of hardening that need to be described. The non-reactive binder generally hardens by drying. However, the reactive binder typically first requires bonding and then hardening. The interfacial forces between the polymer binder and the active materials are shown in Figure 6c. The non-reactive binder mainly relies on intermolecular interactions (Van der Waals force), while the reactive binder mainly relies on hydrogen bonds, covalent bonds, etc. to form interfacial forces. The polymer binder in the electrode made by wet process will perform three different functions, depending on the position of the binder relative to the active material after hardening on the current collector. The binder takes the active material as the center from the inside to the outside, through three layers: a coating modification layer (corresponding to bonded polymer), a fixed connection layer (corresponding to fixed polymer), and a reinforcement stabilization layer (corresponding to excessive polymer). The coating modification layer serves as a modification/covering of the active material, in which the binder covers and modifies the active material’s surface through mechanical interlocking and interfacial reactions; the fixed connection layer mainly serves to connect the active-substance particles; and the reinforcement stabilization layer mainly serves to improve the overall stability of the electrode’s structure. The adhesive strength of the binder relative to the active material depends on the strength of the interaction-force between the coated polymer and the surface of the active particles. The stronger the interaction between them, the higher the adhesive strength of the binder and the higher the limiting ability reducing the volume-expansion and contraction of Si particles. Additionally, the fixed connection and reinforcement stabilization layers polymer characteristics are the major determinant of the binder mechanical properties.

Since the Si particles will produce large amounts of strain due to volume expansion during the Li^+^ insertion and de-insertion, the Si anode binder needs to have not only stronger adhesive strength but also excellent mechanical strength and other properties. According to the structure of polymer binders, researchers have studied a great deal of emerging binders, which may be classified as linear binders, branched binders, crosslinked binders, and conjugated binders. However, in this paper, the polymer binder is separated into two primary types of binders that rely on physical contact (intermolecular forces) and chemical interaction (hydrogen and chemical bonds). This paper summarizes the existing binders based on the strength of the interfacial force between the binder and the Si anode.

### 4.2. Physical Interaction between Binder and Si Anode Surface

The physical interaction between the non-reactive binder and the Si anode mainly relies on the Van der Waals forces, resulting in a relatively weak interfacial force. In the Li^+^ insertion process, the Si particles will break free from the binder due to the massive Si volume expansion. In the Li^+^ de-insertion process, the Si volume shrinks dramatically. However, due to the relatively weak interfacial force of non-reactive binders, the effective bonding state between the active material and the binder is usually not able to be reconstructed, which leads to the separation of the active material from the conductive agent and the current collector. Eventually, the battery’s performance is significantly reduced.

Polyvinylidene fluoride (PVDF) is the most commonly utilized binder in commercial LIBs due to its stable chemical stability, excellent electrochemical properties, and other characteristics [34,49]. PVDF is a non-polar chain polymer material, which requires that one choose N-methyl-2-pyrolidone (NMP) as the solvent, so PVDF is classified as an oil-soluble binder.

PVDF is primarily utilized as the binder in the manufacture of batteries, where the interaction between PVDF and active materials mainly depends on the Van der Waals force. However, when PVDF is applied to a Si anode with significant volume variations, the electrode structure eventually disintegrates because of PVDF’s inapposite mechanical properties and low adhesive force. To further improve both the adhesive strength of PVDF within the Si-based anode and its mechanical properties, Chen et al. prepared a crosslinked poly (vinylidene fluoride-tetrafluoroethylene-propylene) (PVDF-TFE-P) binder with a 160% strain at break, which can effectively cope with the volume expansion problem of Si_0.64_Sn_0.36_ electrode [50], so that the electrode can maintain a specific capacity of 800 mAh g^−1^ over 30 cycles. In addition, Wang et al. copolymerized TFE and VDF in suspension (Figure 7a) to obtain a block copolymer PVDF-b-PTFE binder with high elasticity [51]. The TFE chain segment in this binder can act as a rigid structure to stabilize the deformation of the VDF chain segment by hydrogen bonding with the Si surface (Figure 7b), and its elongation at break can reach 250%. The use of 5 wt% of the block copolymer binder in the Si anode enables the achievement of a high cycling stability with a specific capacity up to 1000 mAh g^−1^ after 250 cycles and a coulombic efficiency of 71.7%, effectively mitigating the volume expansion of silicon and keeping the crushed Si particles together to maintain the stability of the electrode’s structure.

In summary, it is difficult to significantly improve battery performance for Si materials with large volume changes using a polymer binder that only depends on the Van der Waals force as the main adhesive force. Therefore, improving the adhesive force of polymer binders has become the focus of research. (The properties of different Si-based anode binders are summarized in Table 2).

### 4.3. Chemical Bonding of Binder to Si Anode Surface

Research efforts have shown that the oxide layer on the surface of Si particles has Si-O bonds and Si-OH bonds. If the chain structure of the polymer binder contains a large number of polar groups, such as -COOH, -NH_2_, -OH, etc., it can form hydrogen or other covalent bonds with the surface of the Si particles to improve the adhesive force between the binder and the active particles, achieving the purpose of keeping the electrode structure stable [73,74]. This type of binder is often referred to as a reactive binder. The three-dimensional (3D) network structure formed by the reactive binder can better maintain the stability of the silicon anode structure in the charging and discharging process of the battery. Therefore, the use of the reactive binder can improve the capacity of the battery, prolong the service life of the battery, and prevent the phenomena of spontaneous combustion or explosion of the battery.

#### 4.3.1. Polyacrylic Acid (PAA) Binder

The water-soluble binder polyacrylic acid (PAA), which is produced by the polymerization of acrylic acid, offers the advantages of adjustable molecular weight and exceptional mechanical characteristics. Compared with the molecular structure of carboxymethylcellulose (CMC), PAA has a higher concentration of carboxyl groups in the chain segment and a more uniform distribution [75].

In 2010, Magasinski et al. first proposed and used PAA as a Si-based anode binder and found that PAA could form powerful hydrogen bonding on the surface of the anode material and provide a more uniform coating of the anode material [76]. The following year, Komaba et al. conducted a comparative study of three Si anode polymer binders, namely, PAA, CMC, and PVDF [77], and found through a series of experimental tests that the PAA binder could tightly adhere to the anode material, even after several cycles, and effectively inhibit electrolyte deterioration on the anode material, acting as an artificial SEI layer, and finally making the electrode exhibit excellent electrochemical performance. (The Si/C composite electrode using PAA binder exhibits a specific capacity of over 800 mAh g^−1^ after 30 cycles at 1 A g^−1^ and capacity retention of 88.3%).

Although PAA has obvious advantages in terms of adhesion, it was found that the linear structure of PAA is unable to withstand the volume expansion of the Si anode for long durations, and the repeated expansion and contraction of the anode structure will cause the linear binder molecules to slip, eventually leading to the disintegration of the electrode’s structure [53,54,55,69,78]. As a result, scientists began to modify the PAA binder to increase its adhesive strength and mechanical capabilities. These modifications included chemical grafting/crosslinking, as well as physical compounding, which increased the stability of the Si anode’s structure. Inspired by the research on dopamine in mussel foot proteins [52], researchers introduced catechol functional groups into the side chain of PAA in 2013 to obtain a Si-PAA-C binder with extraordinary wetness-resistant adhesion capability in order to enhance the interaction between the PAA binder and the silicon surface (as shown in Figure 8). The results show that the Si-PAA-C binder forms strong hydrogen bonds with Si particles through catechol interactions and significantly improves the cycling performance of Si anode: after 400 cycles; the Si anode employing Si-PAA-C binder was 220 mAh g^−1^ higher in specific capacity at a current density of 2.1 A g^−1^ than the unmodified PAA binder.

In 2014, Song et al. mixed PAA with polyvinyl alcohol (PVA) to create a new water-soluble polymer PAA-PVA binder with a cross-linked network structure by an in-situ crosslinking technique that was applied to the Si anode (Figure 9a) [53]. PVA contains a significant number of hydroxyl groups (-OH) that can form hydrogen bonds with the surface of the Si. On the one hand, the carboxyl functional group (-COOH) of PAA and the hydroxyl functional group (-OH) of PVA will undergo esterification to form a network structure. On the other hand, the carboxylic acid group of PAA will also react with Si-OH bonds and form covalent ester bonds between the silicon’s surface and the polymer binder, and there is also an abundance of hydroxyl groups (-OH) in PVA that can form hydrogen bonds with the silicon [79,80,81], so the PAA-PVA binder can firmly bond with the Si particles and exhibit excellent cycling performance. In 2015, Lim et al. developed a novel physically cross-linked binder PAA-PBI (polybenzimidazole) for application in Si anodes based on reversible acid–base interaction [54], in which basic PBI and acidic PAA interact with each other and produce reversibly-constructed ionic bonds (Figure 9b). The PAA-PBI binder effectively maintained the stability of the Si electrode during expansion and contraction.

In addition to the common cross-linking structures, researchers have explored typical structures according to the needs of practical applications. In 2017 Choi et al. used a molecular pulley design to provide a network topology structure for the PR-PAA binder [55], as shown in Figure 10. The polymer binder is characterized by the formation of sliding cross-linked nodes between polyrotaxanes and PAA, and the α-cyclodextrin(α-CD) is connected to PAA by covalent bonds. The α-CD in the binder slides along the polyethylene glycol (PEG) backbone as the silicon volume expands and contracts during battery charging and discharging. The composite binder is extremely elastic, with a strain of 390%, a level which can withstand huge volume variations in the Si anode and maintain the electrode’s structural stability. The initial specific capacity of a Si anode using this binder is 2971 mAh g^−1^, and the average coulombic efficiency is 99.64% in 23~150 cycles.

In 2015, Wei et al. prepared a NaPAA-g-CMC binder by applying PAA and CMC together in the Si anode [56], utilizing acrylic acid (AA) and CMC as precursors and (NH_4_)_2_S_2_O_8_/NaHSO_3_ as an initiator to create the NaPAA-g-CMC graft copolymer (Figure 11a). The Si anode based on this binder had an excellent stability, with a specific capacity of 1816 mAh g^−1^ after 100 cycles, approximately 79.3% capacity retention, and an average coulomb efficiency of 98.4% within 1–100 cycles. In 2019, Yu et al. reported a CMC-NaPAA-PAM binder with CMC as the backbone, and acrylamide (AM) and acrylic acid as the branched chains [82]. Figure 11b depicts the reaction and the structure of the CMC-NaPAA-PAM binder molecules, with ammonium persulfate as an initiator, and N, N-methylenebisacrylamide (MBAA) as a crosslinker used to produce a three-dimensional hyperbranched polymer binder. Increasing the polar groups (carboxyl and amino) aids in the formation of hydrogen bonds with the Si surface and increases the stability of the anode’s structure and long-term cycle stability.

In 2021 researchers copolymerized lithium acrylate (AALi) and vinyl triethoxy silane (VTEO) in water to obtain a functional siloxane-containing aqueous binder (PAA-VTEO) [57] which hydrolyzed the VTEO groups in the binder under weak acid–base conditions to generate Si-OH groups. The Si-OH groups readily interacted with Si particles and the copolymer binder to produce Si-OH bonds in order to form Si-O-Si chemical bonds. These Si-O-Si covalent bonds facilitated the construction of a 3D cross-linked network structure and increased the intermolecular forces (Figure 12a), which effectively suppressed the silicon’s volume-expansion effect and significantly enhanced the cycling stability of the C@Si470 electrode (with a capacity retention of 96.7% after 200 cycles).

In 2022, Liu et al. successfully constructed a soft-rigid double chain network structure PAA-SS binder by, in situ, cross-linking PAA and sodium silicate (SS) [33] (Figure 12b); the soft chain of PAA and the rigid chain of SS synergistically enhanced the adhesion force and mechanical strength of the binder. The Si electrode with PAA-SS binder had a high initial coulombic efficiency of 93.2% and an 88.2% capacity retention after 500 cycles. In the same year, Pan et al. used glycinamide hydrochloride (GA) to modify PAA to obtain PAG (Figure 12c). PAG is cross-linked with epoxidized natural rubber (ENR) to obtain PE55 binder (PAG:ENR = 1:1 cross-linked polymer) [58], which has better adhesion force (4.45 N). Its adhesion force with Si particles and the current collector can effectively maintain the stability of the electrode’s structure and ensure the normal operation of the electrode.

#### 4.3.2. Sodium Carboxymethyl Cellulose (CMC) Binder

Sodium carboxymethyl cellulose is a linear cellulose derivative in which -CH_2_COONa replaces H in the -OH group on the glucose ring of the cellulose monomer. The carboxymethyl substitution degree in CMC can be adjusted. Its maximum degree of substitution is three, and the three carboxyl groups in each C_6_ ring can be dissociated into -COO^−^ anion [83].

In 2006, Buqa et al. compared the cycling ability of electrodes using CMC binder with styrene butadiene rubber (SBR), CMC binder, and PVDF binder using C/Si composites as the anode active material [84]. Their findings showed that electrodes using 1% CMC binder containing 1% SBR had the same cycling stability as the same electrode material using 10% PVDF binder. Currently, the CMC-SBR binder is the most widely used commercial water-soluble binder [46]. In 2008, Hochgatterer et al. conducted additional research on CMC binders, postulating a covalent ester-like bond mechanism, and discovered that CMC with a high degree of carboxymethyl substitution was more advantageous to the cycling performance of Si-based anodes [85]. They proposed that the -CH_2_COONa group in CMC hydrolyzes first to generate -CH_2_COOH, and then establishes an ester bond contact with the -OH group on the surface of the Si, which strengthens the adhesive force between the binder and the surface of the active-material silicon. Based on the above mechanism of covalent ester-like bond, Mazouzi et al. showed that Si-OH groups on the Si surface and -COOH groups in CMC can have stronger chemical interaction when pH = 3, further ensuring better adhesion between the binder and the active substance [86]. The following year, Bridel et al. proposed a hydrogen bonding mechanism for CMC [87], contending that the -CH_2_COOH group produced by the hydrolysis of the -CH_2_COONa group forms a hydrogen bonding interaction with the -OH group on the silicon’s surface with a self-repairing capacity, and that this hydrogen bond self-repairing capacity can accommodate the active material’s larger volume expansion. In addition, Munao et al. analyzed CMC and Si materials (using CMC binder) by their infrared spectra and found that the absorption peaks of carboxyl groups were slightly shifted and no ester bond absorption peaks appeared, which further proved the theory of the hydrogen bond mechanism [88]. However, either mechanism can prove that the presence of -COOH in the binder can improve the adhesive force of the binder and effectively ensure the stability of the electrode’s structure.

To increase the cycling performance of Si anodes, the researchers subsequently changed the CMC binder via grafting and cross-linking. In 2017, Liu et al. prepared a CMC-CA cross-linked binder [59] (Figure 13), in which the carboxyl groups in citric acid (CA) and the hydroxyl groups in CMC underwent a condensation reaction at 150 °C under vacuum and formed ester groups through intermolecular chain cross-linking to enhance the mechanical properties of the composite binder. The linear structure CMC cross-linked into the 3D network structure can reduce the slip of binder molecules due to Si volume-expansion. In addition, the three-dimensional network structure of the binder helps to bond the active material, conductive agent, and current collector together to maintain the stability of the electrode’s structure. The active material of the CMC-CA cross-linked binder is double-shelled yolk-structured silicon (CVSS). Other conditions remained unchanged, and only the binder type was changed. The result shows that the capacity of the CVSS electrode after 100 cycles with the cross-linked CMC binder is about twice that of the CMC binder, and coulomb efficiency stabilizes above 99% after eight cycles. Additionally, at a high current density of 5 A g^−1^, the CVSS electrode capacity retention is 84.6%, even after 1000 cycles.

In 2019, Lee et al. copolymerized CMC with polyethylene glycol (PEG) to obtain the CMC-PEG binder [60] (Figure 14a), which is rich in carboxyl, ether, and hydroxyl groups that can offer some bonding sites which in three dimensions form hydrogen and ester bonds with the active material Si surface to effectively improve the bonding between electrode materials (Figure 14b).

In 2022, Zhang et al. synthesized Si/C composite materials from photovoltaic (PV) waste Si and constructed the 3D cross-linked binder by hydrogen bonding interactions between CMC polymer and ethylenediaminetetraacetic acid disodium (EDTA-2Na), followed by the addition of Ca^2+^ with coordination between EDTA-2Na [31] (Figure 15). The 3D cross-linked CMC/EDTA-Ca^2+^ binder outperformed the CMC binder in terms of mechanical characteristics and adhesion strength.

In the same year, Tang et al. designed an ultralow-content (1 wt%) three-dimensional cross-linked binder (LiCMC-TA) for the first time, which improved electrochemical performance while being able to endure the significant volume change of the silicon anode [61]. The three-dimensional cross-linked LiCMC-TA binder used partially lithiated carboxymethyl cellulose (LiCMC) as the backbone structure and tannic acid (TA) as the cross-linker (Figure 16), which synthesized a three-dimensional cross-linked binder by multiple hydrogen bonding that effectively buffered the stress generated by the volume expansion of silicon. In addition, Li^+^ can be rapidly transferred through some lithiated groups of LiCMC-TA binder to make a Si anode with excellent cycling performance.

#### 4.3.3. Natural Polymer Binder

Natural polymer binders with numerous polar groups (-COOH, -OH) such as sodium alginate (Alg), chitosan (CS), cyclodextrin (β-CDp), guar gum (GG), etc. have strong adhesive force between the silicon’s surface and the current collector. Furthermore, natural binders are plentiful, inexpensive, and environmentally friendly, making them potential Si anode binders.

I.Alginate (Alg) binder

Sodium alginate (Sodium Alginate, SA/Na-Alg) is a natural polysaccharide composed of β-D-mannuronic acid (M) and α-L-guluronic acid (G), one which generally exists in the form of sodium salts. Compared with CMC binder, alginate (Alg) chain has a more uniform distribution of carboxyl groups.

In 2014, Liu et al. explored the Ca-Alg binder with the structure shown in Figure 17a. Unlike the previous binders functioning through covalent cross-linking, ionic dynamic cross-linking of Alg can be performed through the G blocks chain, which makes the binder self-healing, and maintains the stability of the electrode’s structure [62]. Adding a small amount of CaCl_2_ to Alg forms a cross-linked Ca-Alg binder. The ionic bond formed between Ca^2+^ ions and carboxyl groups in Alg causes the Alg molecular chains to rearrange, depending on the cross-linking effect. More amorphous structures and better electrochemical properties are obtained in the end.

In 2017, Wu et al. formed alginate molecule chains that were coordinated with various cations to create different alginate hydrogel binders (M-alg, M = Al, Ba, Mn, and Zn) (Figure 17b) [63], which could then be used for Si anodes to improve their stability. The Vickers hardness of both Al-alg and Ba-alg is higher than 32.0 HV, while the Vickers hardness of the remaining M-alg is less than 25.5 HV. In addition, the viscosity of Ba-alg binder is 30270 mPa s, while the viscosity of the other M-alg binders is less than 1700 mPa-s. The Si anode with Al-alg or Ba-alg binder has a specific capacity that is close to 2100 mAh g^−1^ after 300 cycles at 0.42 A g^−1^. In 2022, Zhu et al. obtained PAA-SA binder by moderate cross-linking of PAA with sodium alginate. The discharge capacity of Si anode using PAA-SA binder was 1419.8 mAh g^−1^ in 1 A g^−1^ after 200 cycles, which is much higher than sodium alginate binder, and coulomb efficiency of 99.5% in 100 cycles [30].

II.Chitosan (CS) binder

Chitosan (CS) is the second-most-common biopolymer in the world after cellulose. It is found primarily in the shells of arthropods such as shrimp, crabs, and shell worms. Chitosan is deacetylated by chitin and then carboxylated to form a water-soluble binder application for Si anodes [89]. The specific synthesis process and bonding mechanism of chitosan with Si particles are shown in Figure 18a.

In 2018, researchers obtained epoxidized natural rubber (ENR) by the conventional peroxyacetic acid process and obtained CS/ENR binder with high adhesion and elastic by cross-linking [64] (Figure 18b). The chitosan can firmly hold the active material silicon particles by hydrogen bonding due to the existence of hydroxyl and amino groups. Besides, the rubber acts as the buffer layer that reversibly stretches and shrinks during the alloying/dealloying, preventing the shedding of the active material. The CS/ENR binder keeps the electrode structure stable and gives the electrode outstanding cycling performance. The specific capacity of the electrode prepared with this binder, after 500 cycles, is 2310 mAh g^−1^ with a capacity retention of 87.8% at 1 A g^−1^. Additionally, the electrode still has a high specific capacity of 1350 mAh g^−1^ after 1600 cycles at a high current density of 8 A g^−1^. In 2022, Liao et al. created a 3D dynamic cross-linked binder (CS-EDTA) for SiO_x_ electrodes [65]. The reversible ionic bonding of the CS-EDTA binder forms the dynamic cross-linked structure that can be repaired at the fracture point of the electrode. The dynamic cross-linked structure relieves the volume expansion of the SiO_x_ material and enhances the stability of the SiO_x_ anode structure.

III.β-cyclodextrin polymer (β-CDp)-based binders

β-cyclodextrin (β-CD) is a macrocyclic oligosaccharide compound with seven (1,4)-R-D-glucopyranose units, one easily synthesized by enzymatic degradation of starch and further functionalized with epichlorohydrin (EPI) under strong alkaline conditions to form hyperbranched β-cyclodextrin polymer (β-CDp) with polyhydroxy groups (Figure 19a). The hyperbranched structure of β-CDp exhibits multidimensional hydrogen bond interactions with silicon, providing a strong contact site that resists the mechanical stress generated by silicon expansion, and forms a stable electrode [90] (Figure 19a).

In 2015, Kwon et al. formed hyperbranched β-cyclodextrin polymers via a supramolecular cross-linker made from dendritic gallic acid that contained six adamantane (AD) units [66]. The specific process is shown in Figure 19b; the β-CDp/6AD binder reversibly restores the broken connections between polymer chains during the electrode’s continuous cycle through dynamic host–guest interactions. The dynamic crosslinking process achieves close interaction of the β-CDp/6AD binder and improves the cycling performance of the Si anode (90% capacity retention about 1600 mAh g^−1^ after 150 cycles at 1.5 A g^−1^).

In 2022, Zheng et al. used linear PAA and partially neutralized sodium polyacrylate (PAAS) as the binder framework, and esterified β-CDp with the carboxyl groups on the framework to form a stable chemical bond (ester, ionic) and hydrogen bonds to synergistically design a three-dimensional cross-linked network structure to obtain PAAS-β-CDp-PAA binder [39] (Figure 20a). This PAAS-β-CDp-PAA binder can enhance the adhesion between the active substance and the current collector. In addition, the unbonded carboxyl and hydroxyl groups on this binder can make complete contact with the hydroxyl groups on the surface of the Si particles to form hydrogen bonds. The hydrogen bond can effectively alleviate the expansion and shedding of the active material Si (Figure 20b), promoting the stability of the SEI layer and maintaining the integrity of the electrode.

IV.Guar Gum (GG) binder

Guar Gum (GG) is a kind of galactomannan extracted from the leguminous plant, guar bean. There are a large number of hydroxyl groups in its molecular structure that can form hydrogen bonds with the surface of active particles of silicon and form strong interaction forces with the silicon particles. In addition, Liu et al. discovered that GG is comparable to the solid polymer electrolyte polyethylene oxide (PEO) and can effectively transfer lithium-ions to enhance the electrode’s ionic conductivity [91], as shown in Figure 21a.

In 2022, Li et al. prepared a binder with high ionic conductivity (GG-g-PAM) using polyacrylamide (PAM) combined with guar gum backbone [45], as shown in Figure 21b. The lithium complexation sites provided by the oxygen heteroatom in GG constructed the Li^+^ transport channel in the Si anode, resulting in a good cyclability of the Si anode. The specific capacity of the electrode is maintained at 1687 mAh g^−1^ after 200 cycles in 1 A g^−1^ with a capacity retention of 71.8%. In the same year, Tong et al. prepared a GCA13 binder by introducing citric acid (CA) molecules onto the long chain of GG [36] (Figure 21c); the short-range effect exerted by CA can buffer the volume change to reduce the Si particle-crushing phenomenon. Moreover, due to the excellent performance of the GCA13 binder, the Si-based anode with GCA13 can be applied under harsh conditions. The electrode exhibits remarkable cycling stability at low (−15 and 0 °C) and high temperatures (60 °C). The electrode provides a stable reversible capacity of 1025 mAh g^−1^ and 1578 mAh g^−1^ after 200 cycles at 15 °C and 0 °C, respectively; the electrode maintains a reversible capacity of 2302 mAh g^−1^ after 55 cycles at 60 °C.

V.Other natural binders

Xanthan gum (XG) is similar to millipedes in structure. Its main chain is the same as cellulose. The side chain is wound backward around the main chain and forms a double-helix structure through hydrogen bonding, as shown in Figure 22a. The charged groups located on the trisaccharide side chains are not conducive to this entangling process, though, due to coulomb repulsion, which causes the side chains to partially detach from the helical backbone. The mannose residues attach to the backbone to be acetylated, allowing the terminal groups of the helical backbone to promote the structure for self-assembly and interact with the adjacent backbone. According to the mechanism, the helical structure continues to form in the two different backbones [92], as shown in Figure 22b, and, finally, the XG binder with double helical structure is more stable than the CMC and Alg binders.

Furthermore, in 2020 Choy et al. first reported that natural spider silk (SWS) can effectively maintain the electrode capacity and reduce the volume expansion of the Si anode in LIBs [93]. The excellent electrochemical performance and recycling ability of the electrode with SWS may be attributed to the unique properties of SWS, such as the folded crystal layer with semi-amorphous structure, appropriate viscosity, adhesion properties, tight stacking of protein blocks and side chain groups. Natural sesbania gum (SG) was first used in 2022 by Wang et al. as an effective biopolymer binder for commercial SiO_x_/C composite anodes and Si anodes [37]. SG has a branched structure and an abundance of polar groups (Figure 22c) with high mechanical properties and adhesion. A Si electrode with SG has a high initial coulombic efficiency of 90.92% and a capacity of 2023 mAh g^−1^ after 120 cycles at a current density of 1 A g^−1^. Furthermore, the SG binder applies to commercial SiO_x_/C composite anode that, mixed with LA136D binder (composed of acrylic acid derivative polymer), shows excellent synergistic effect. The SiO_x_/C composite electrode shows the capacity retention exceeded 92% after 100 cycles. In the same year, Ma et al. applied the binder prepared with polyurethane (PU) and polydopamine (PDA) to the silicon anode, which showed excellent cycling stability (with a capacity about 1000 mAh/g after 1000 cycles). The polyurethane has outstanding mechanical properties and the polydopamine has strong adhesion effect, promoting the stable performance of the electrode [94].

#### 4.3.4. Functional Binder

The main function of Si-based anode binders is to alleviate the expansion of anode materials and maintain the stability of the Si anode’s structure. In addition to the most basic adhesion ability, the development of some new binder systems and their self-healing and conductivity can substantially enhance the electrochemical performance and cycle-stability of the Si anode [95].

I.Self-healing binder

Dynamic crosslinking can enable the binder to have self-healing performance. In 2018, Yang et al. synthesized the UPy-PEG-UPy binder with quadruple hydrogen bonds [67] (Figure 23a). The Si anode, using this binder, has good electrochemical performance, with a reversible capacity of 1454 mAh g^−1^ after 400 cycles and the capacity retention of 84%. The hydrogen bond with high strength in the molecule can give binder a self-healing function. The excellent “self-healing effect” can repair the small cracks of the Si anode and maintain the integrity of the anode structure during Li^+^ insertion and de-insertion [96]. In addition, in 2020 Jiao et al. designed a double-wrapped PAA-BFPU binder with low Young’s modulus bifunctional polyurethane (BFPU) as a buffer layer to dissipate the stresses generated by volume expansion of the Si anode during cycling [68]. The molecular structure of the binder is shown in Figure 23b; the BFPU structure contains disulfide bonds, which can heal the tiny cracks produced by the Si anode in a short time and prolong the battery’s lifespan. The specific capacity of the Si electrode with the binder is about 3000 mAh g^−1^ at 1.2 A g^−1^ after 100 cycles with a capacity retention of 97.0%. In 2022, Malik et al. designed a self-healing binder PDPP (polydioxythiophene: polyacrylic acid: phytic acid, PEDOT:PAA:PA) with high tensile properties that can self-repair cracks and damage generated in electrodes during cycling [38]. For the first time, it was demonstrated that the artificially created cracks in the electrolyte of Si anode could heal automatically under real battery operating conditions (Figure 23c). The Si anode using the PDPP binder provided a reversible capacity of 2312 mAh g^−1^ after 100 cycles with an initial coulombic efficiency of 94%. Besides, the self-healing effect of the Si anode was quantified for the first time in this paper. The self-healing efficiency of Si electrodes was obtained by profilometry (Profile), electrochemical impedance spectroscopy (SEI), and four-point probe (4PP), three different methods (Figure 23d).

II.Conductive binder

Although Si as the anode material for LIBs has a high theoretical specific capacity, the primary problems to solve are the volume expansion of Si during the alloying/de-alloying, followed by improving the electrical conductivity of the Si electrode. As shown in Figure 24A, the traditional electrode material is a three-part system made up of the active chemical, the conductive agent, and the polymer binder [6]. Although the polymer binders and conductive agent are necessary, the addition of large amounts of inactive materials can reduce the energy density of the battery. Additionally, the normal conductive agent easily loses touch with the electrode’s active material during the Li^+^ insertion and de-insertion, which results in the failure of electrode conductivity and a quick decline in electrode capacity. As shown in Figure 24B, the use of conductive binder as both conductive agent and binder for the Si anode can decrease the quantity of inactive material applied in the electrode material, improve the loading of the electrode active material, and provide a conductive network structure to ensure the smooth flow of electron transport channels during the cycle of the battery, ultimately increasing the electrode’s capacity.

Some binders commonly used in scientific research, such as PAA, CMC, Alg, and CS, do not have conductivity. Researchers have doped the polymers to make them conductive in order to increase the binder conductivity. In 2013, Wu et al. created a Si anode with polyaniline (PANI) gel as a binder using phytic acid as a cross-linker by an in-situ polymerization method [97]. A series of electrochemical tests showed that the silicon anode prepared by PANI gel has higher specific capacity and stability than that prepared by PVDF or the common PANI binder. The silicon anode with PANI gel maintains a specific capacity of about 1600 mAh g^−1^ after 1000 cycles at 1 A g^−1^. It also has outstanding cycling performance under a high current density of 6 A g^−1^, and the capacity retention is still above 90% about 550 mAh g^−1^ after 5000 cycles. In 2016, Lee et al. modulated the properties of PAA by incorporating PANI components; the addition of PANI can improve the electronic conductivity of the binder [69]. In addition, the amino group associated with PANI reversibly interacts with the carboxylic acid on PAA through an acid-base reaction which can improve the mechanical properties of the polymer binder. The test results demonstrate that the initial discharge capacity (1979 mAh g^−1^) and the capacity retention (56.5%) after 300 cycles are higher for the LIBs with a PAA binder containing 10% PANI than for the LIBs without PANI binder (initial discharge capacity 1913 mAh g^−1^, capacity retention 11.3%). In the same year, Higgins et al. used a conducting polymer poly (3,4-ethylenedioxythiophene)/poly-(styrene-4-sulfonate) (PEDOT:PSS) as a binder for the Si anode [70], eliminating the capacity loss from the separation of Si and inorganic conducting additives during the Li^+^ insertion and de-insertion. (After 100 cycles of the electrode at a current density of 1 A g^−1^ discharge capacity retention reached 75% of 1950 mAh g^−1^). The highly stretchable conductive binder (CG) was obtained by adding D-sorbitol and VAA (polyvinyl acetate-acrylate) basis on PEDOT:PSS, which has strong conductivity and tensile properties. The new binder does not lose conductivity when the volume expands by a factor of four, and can maintain the stability of the Si anode’s structure [71]. In 2021, Su et al. cross-linked the stretchable polymer poly(ether-thioureas) (PETU) with (PEDOT:PSS) to synthesize a multifunctional polymer binder (PPTU) [72]. The PPTU binder has high electrical conductivity, high elasticity, and self-healing properties, enabling the Si anode to exhibit strong cycling stability and high current-discharge capability The Si anode with PPTU has a reversible capacity of 2081 mAh g^−1^ at 1 A g^−1^ after 300 cycles. In 2022, Song et al. introduced dopamine-functionalized fluorene structural units (DA) into polyfluorene-type copolymers (PFPQ) [35] (Figure 25a). The copolymers’ adhesive force and conductivity increase to 4.39 N and 2.28 × 10^−2^ S cm^−1^.

Through the interaction between the PFPQDA polymer chain and the active material, the addition of polar groups in the DA group contributes to the construction of three-dimensional networks and elastic electron transporter in the Si and SiO_x_ electrodes. The PFPQDA binder mitigates the expansion of the Si-based anode during long-term cycling and maintains the integrity of the conductive pathways, improving the electrochemical performance (reversible capacity of the Si electrode after 150 cycles at 0.42 A g^−1^ current density is 2523 mAh g^−1^, with a capacity retention of 96%).

### 4.4. Summary

According to the survey, the majority of Si anode binders investigated thus far are green water-soluble binders which primarily rely on chemical bonding with the Si anode’s surface. Polymer binders, typically featuring polar groups like -COOH, -OH and -NH_2_, can increase the binder water solubility and encourage the interaction of hydrogen and covalent bonds [98]. To improve the adhesive force and mechanical qualities of the binder, it is typically necessary to create a three-dimensional network structure. The mechanical properties of the binder are improved, while the stability of the electrode’s structure is maintained [99]. Additionally, the binder can be created within a double-wrapped structure. The inner rigid binder acts as a barrier to prevent the volume expansion of silicon and dissipate its stress. The outer elastic binder acts as a buffer to consume the remaining stress and maintain the stability of the silicon anode structure during the charging and discharging process to ensure that the battery can be used normally. At the same time, it is possible to make the binder conductive and improve the ion transport efficiency of the electrode by proper pretreatment, or by using materials with high lithium-ion transport efficiency. Increasing the hydrogen bonding and disulfide bonding in the binder makes the binder have certain self-healing properties. The above methods can improve the performance of the binder.

## 5. Conclusions and Outlook

Si, as anode material for lithium-ion batteries, has excellent prospects for development, but whether the Si volume expansion problem can be solved is a key factor in the existence of the Si anode. Merely changing the Si materials’ structure and electrolyte additives can’t solve all the problems; the use of binder can ensure the integrity of the electrode’s structure and greatly improve the cycle-stability of the Si anode. In the process of designing the binder, it is necessary to consider (1) more bonding sites between the binder and the Si particles to improve the adhesive force, such as through the introduction of catechol groups to ensure excellent adhesive strength. At the same time, increasing the number of polar groups on the binder, such as -COOH, -OH, -NH_2_, etc. has several effects: (1) It increases the roles of hydrogen bonding and covalent bonding, giving the binder better adhesion properties and better water solubility [100]. (2) It improves the mechanical strength of the binder through grafting or cross-linking the binder to form a three-dimensional network structure. A double-wrapped structure formed through rigid and flexible polymers can prevent the binder from slipping in the process of charge/discharge, so that the Si particles, conductive agents and current collector form better contacts within the electrode’s structure to maintain stability. (3) It gives the Si binder more desirable properties, such as chemical stability and electrochemical stability, so that the binder has self-healing ability, conductivity, and ionic conductivity, etc. [101] Self-healing binders can spontaneously heal the tiny cracks generated by Si expansion and improve the electrode’s cycle stability. Making the binder conductive reduces the amount of conductive agent in the electrode, thereby increasing the loading of active materials. Improving the binder’s ionic conductivity ensures the efficiency of the lithium-ions transport during the battery charger/discharge cycle. In a word, researchers have used a variety of techniques to create binders with outstanding qualities in the Si anode to reduce Si volume expansion, preserve the structural integrity and boost lithium-ion battery capacity [46,73,102,103,104,105].

## Figures and Tables

**Figure 1 materials-16-04266-f001:**
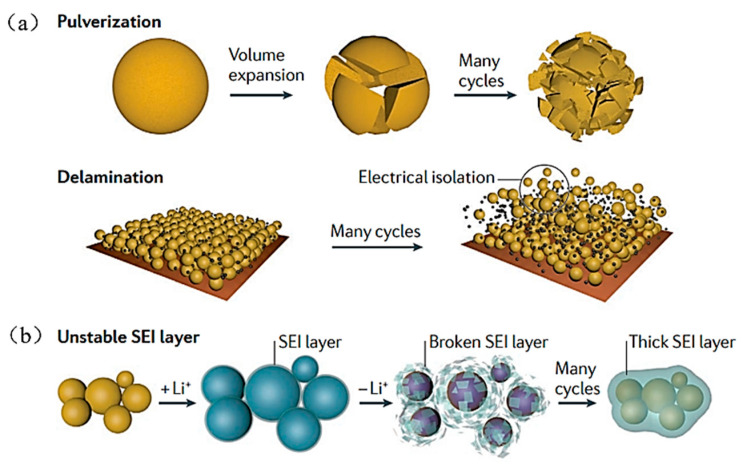
Main degradation mechanisms of a silicon anode (**a**) The pulverization of silicon particles and the delamination of electrodes. (**b**) The change of unstable SEI layer [9].

**Figure 2 materials-16-04266-f002:**
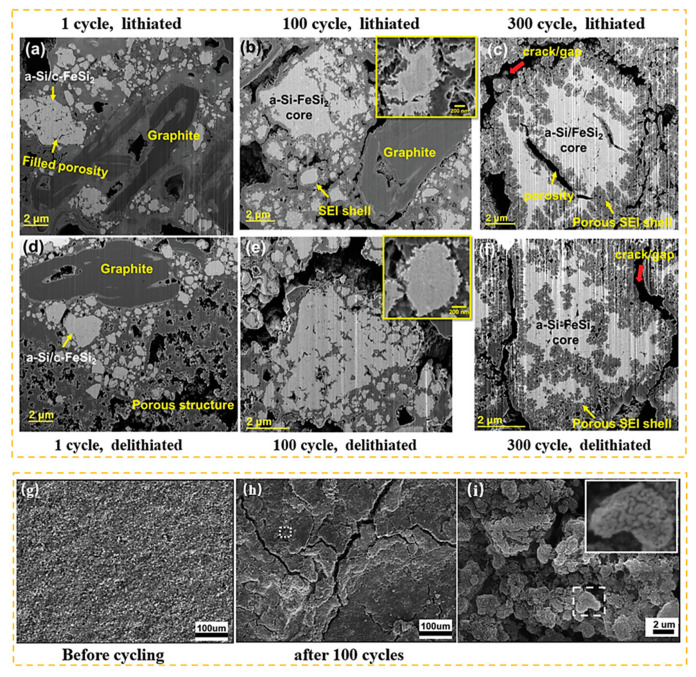
(**a**–**f**) Cross-section of the a-Si/c-FeSi_2_ alloy/graphite electrode after different cycles of lithiation/de-lithiation [12]. (**g**–**i**) SEM images of Si electrode before and after cycling [13].

**Figure 3 materials-16-04266-f003:**
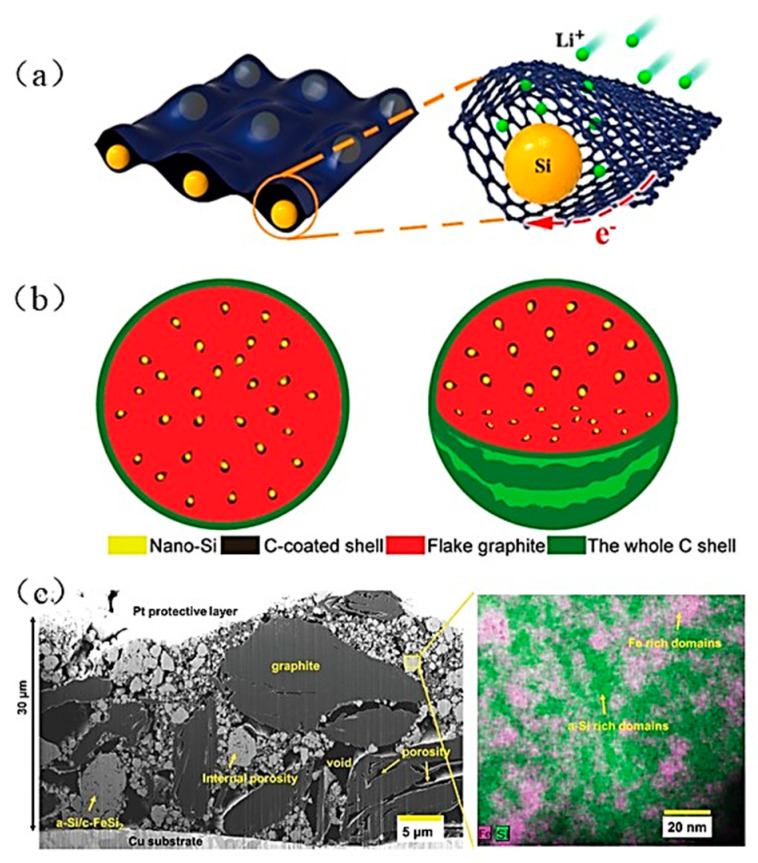
(**a**) Model of silicon nanoparticles wrapped by two-dimensional carbon nanosheets [28]. (**b**) Schematic diagram of Si/C microspheres of watermelon model [29]. (**c**) FIB–SEM image of cross-sectional of the original electrode and associated elemental mapping images of a-Si/c-FeSi_2_ alloy particle [12].

**Figure 4 materials-16-04266-f004:**
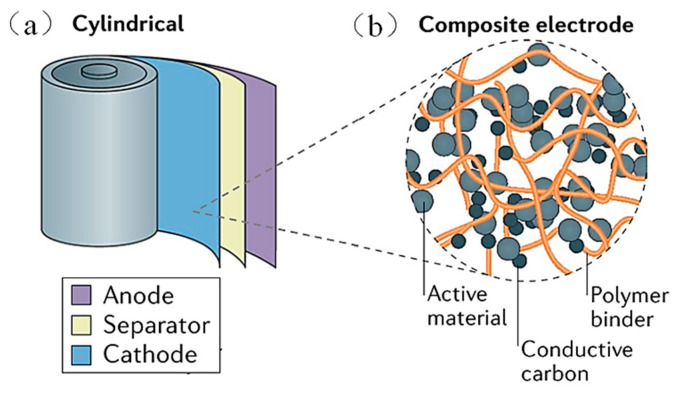
(**a**) Schematic diagram of the cylindrical lithium-ion batteries. (**b**) The electrode consists of active substances, conductive agents, and polymer binders [42].

**Figure 5 materials-16-04266-f005:**
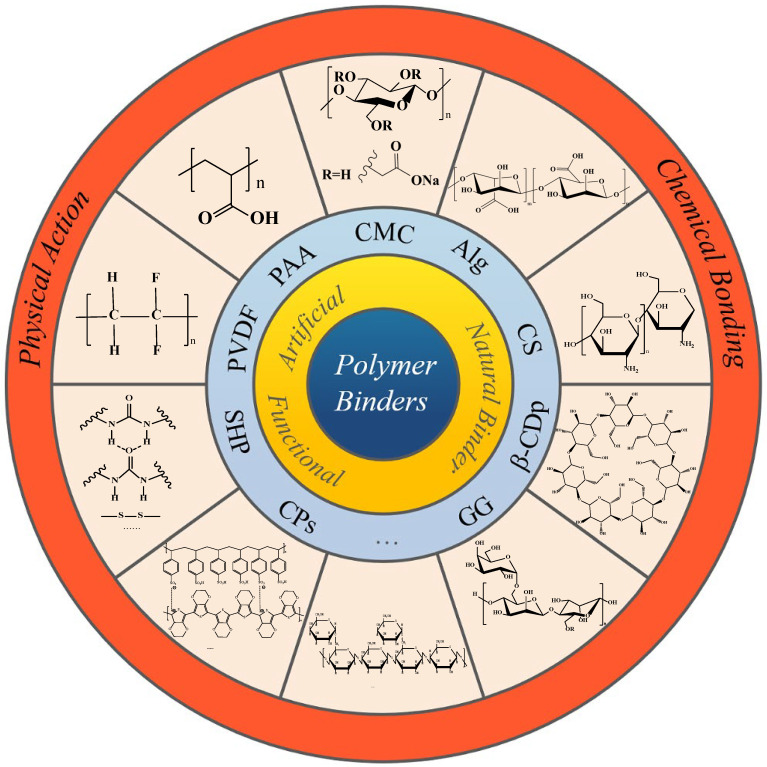
Schematic diagram of the different types of silicon anode binders.

**Figure 6 materials-16-04266-f006:**
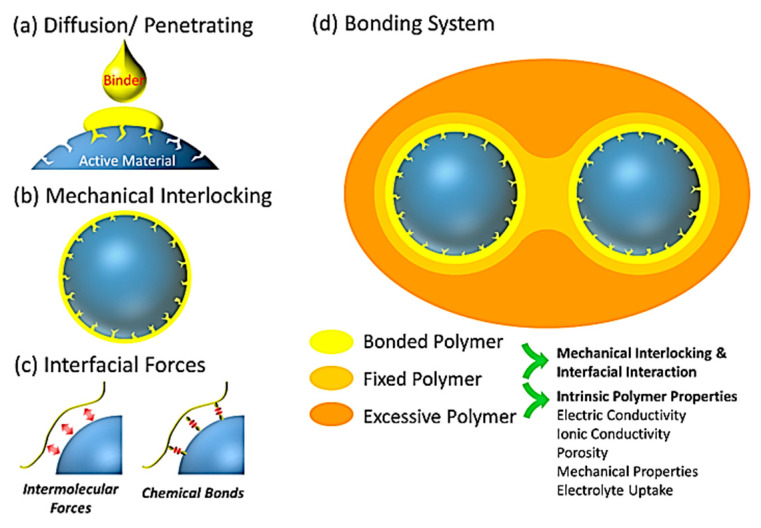
Schematic illustration of the binding mechanism [48].

**Figure 7 materials-16-04266-f007:**
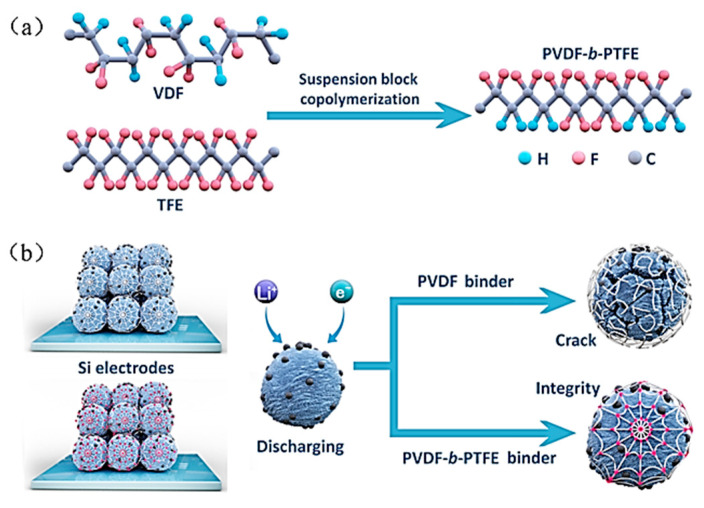
(**a**) Diagram of synthesis of PVDF-b-PTFE binder. (**b**) Schematic diagram of the PVDF-b-PTFE binder stabilizing the Si electrode during alloying/de-alloying [51].

**Figure 8 materials-16-04266-f008:**
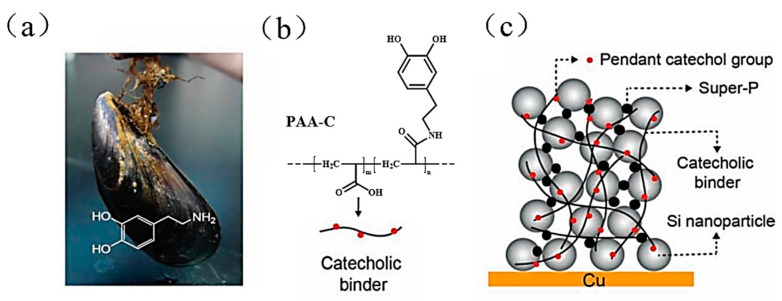
(**a**) Mussel; dopamine structure from mussel foot protein. (**b**) PAA-C binder structural formula and simplified formula. (**c**) Illustration of Si anode structure [52].

**Figure 9 materials-16-04266-f009:**
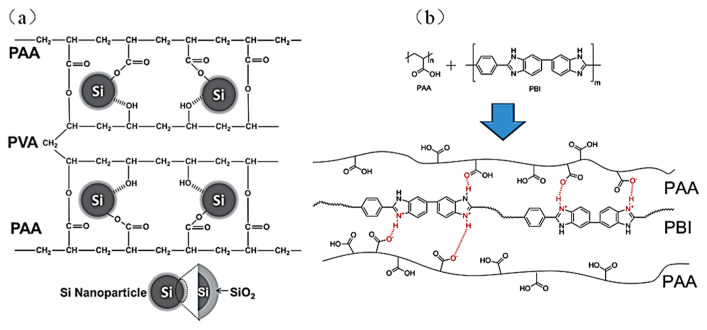
(**a**) Molecular structure and interaction between PAA-PVA binder and Si nanoparticles [53]. (**b**) Schematic diagram of PAA and PBI forming physical crosslinked polymer binders through reversible interactions [54].

**Figure 10 materials-16-04266-f010:**
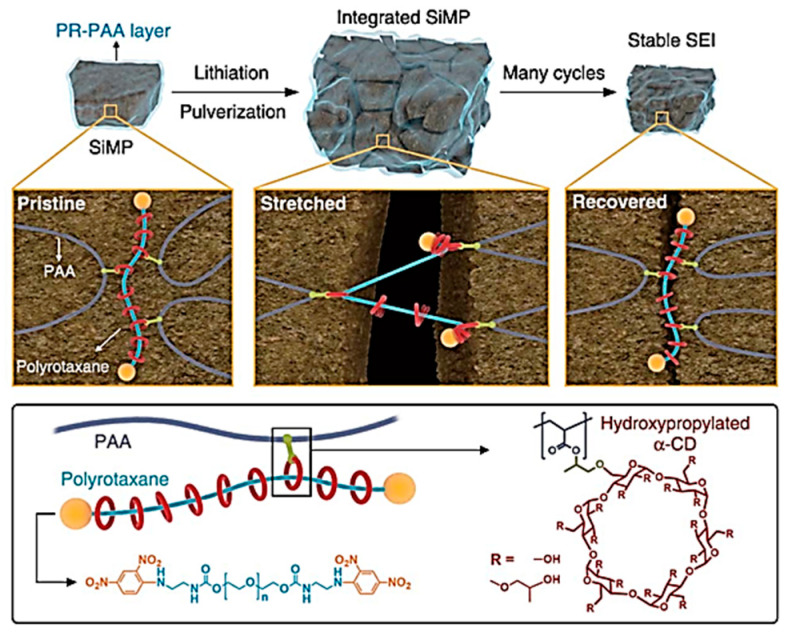
Dispersion stress of PR-PAA binder under repeated change of the volume of Si particles and molecular structure of the PR-PAA binder [55].

**Figure 11 materials-16-04266-f011:**
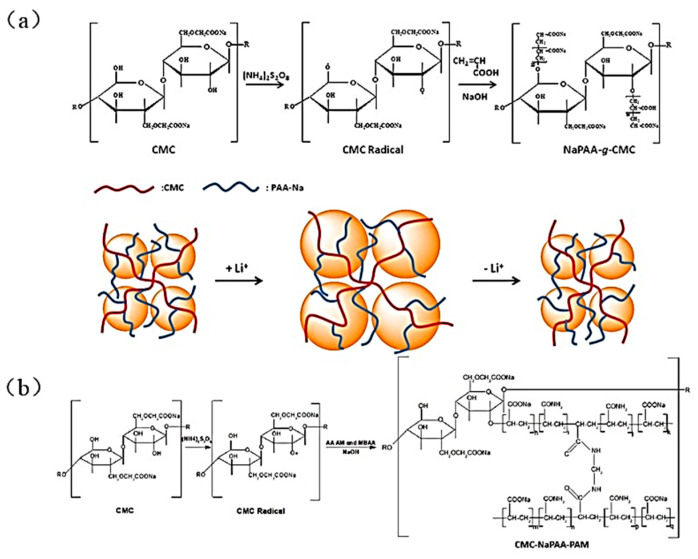
(**a**) Preparation of NaPAA-g-CMC binder and the mechanism of binder adaptation to Si volume change during Li^+^ insertion/de-insertion processes [56]. (**b**) Scheme for the synthesis of CMC-NaPAA-PAM binder [82].

**Figure 12 materials-16-04266-f012:**
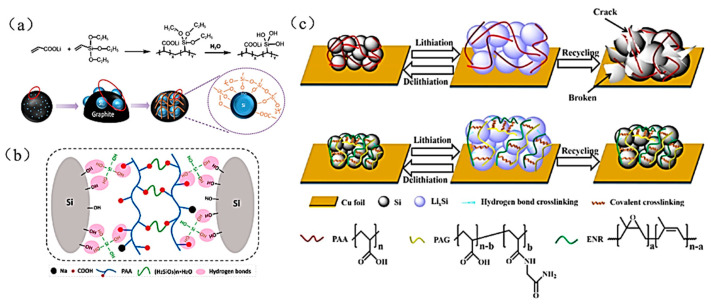
(**a**) Schematic diagram of the adhesion between Si/C electrode and PAA-VTEO binder [57]. (**b**) Schematic diagram of molecular interaction between PAA-SS binder and Si [33]. (**c**) Working mechanism of PAA binder and PAG/ENR binder [58].

**Figure 13 materials-16-04266-f013:**
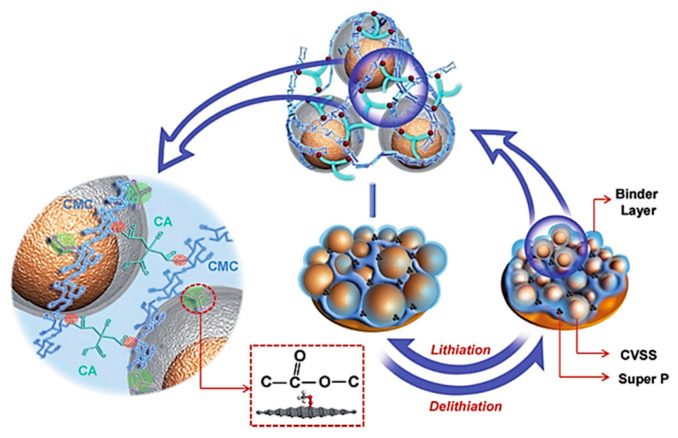
Schematic diagram of the interaction between CVSS and CMC-CA binder [59].

**Figure 14 materials-16-04266-f014:**
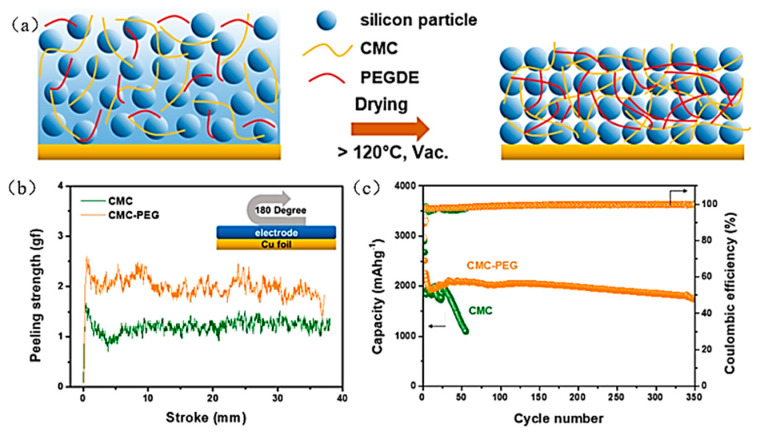
(**a**) Schematic illustration of CMC and PEG in situ cross-linking in Si electrode during drying process. (**b**) A 180° peeling experiment to measure the adhesion and cohesion strength of the Si electrode. (**c**) Cycling performance of silicon electrode [60].

**Figure 15 materials-16-04266-f015:**
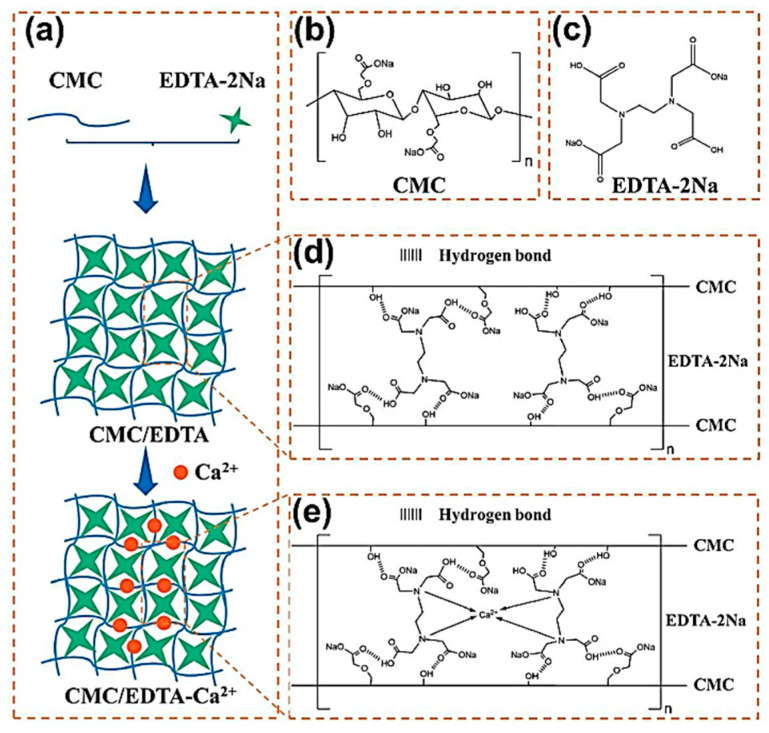
(**a**) Schematic diagrams of cross-linked CMC/EDTA and CMC/EDTA-Ca^2+^ binders. (**b**) Molecular structure of CMC. (**c**) Molecular structure of EDTA-2Na. (**d**) Synthesis of CMC/EDTA binder. (**e**) Synthesis of CMC/EDTA-Ca^2+^ binder [31].

**Figure 16 materials-16-04266-f016:**
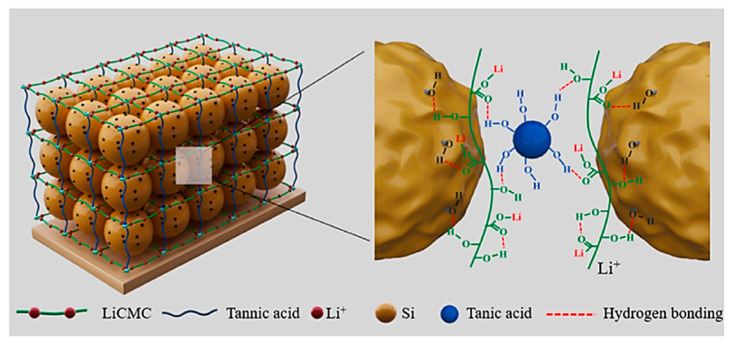
Three-dimensional-crosslinked illustration of the Si anode based on the LiCMC-TA binder [61].

**Figure 17 materials-16-04266-f017:**
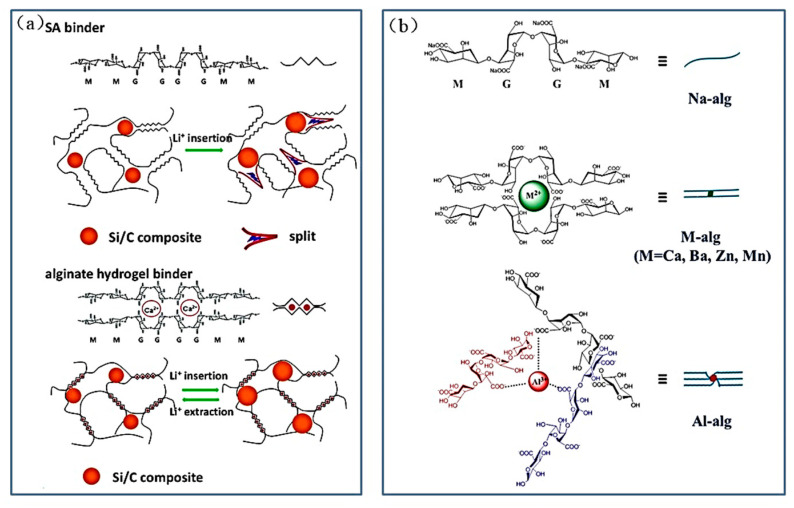
(**a**) Effect of Ca-Alg binder on the structural integrity of the electrode during cycling [62]. (**b**) Schematic diagram of alginate binder with different cationic coordination [63].

**Figure 18 materials-16-04266-f018:**
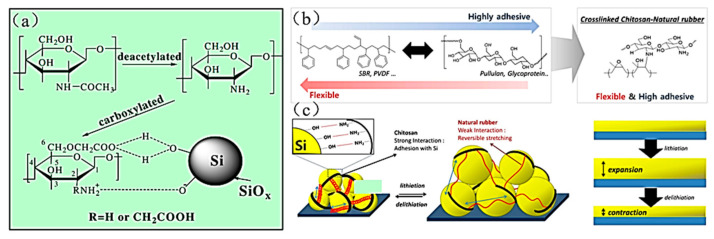
(**a**) Preparation process of chitosan binder and the bonding mechanism with the nano silicon surface [89]. (**b**) Design of CS-ENR binder. (**c**) Schematic diagram of CS-ENR binder mechanism and electrode changes during charging and discharging [64].

**Figure 19 materials-16-04266-f019:**
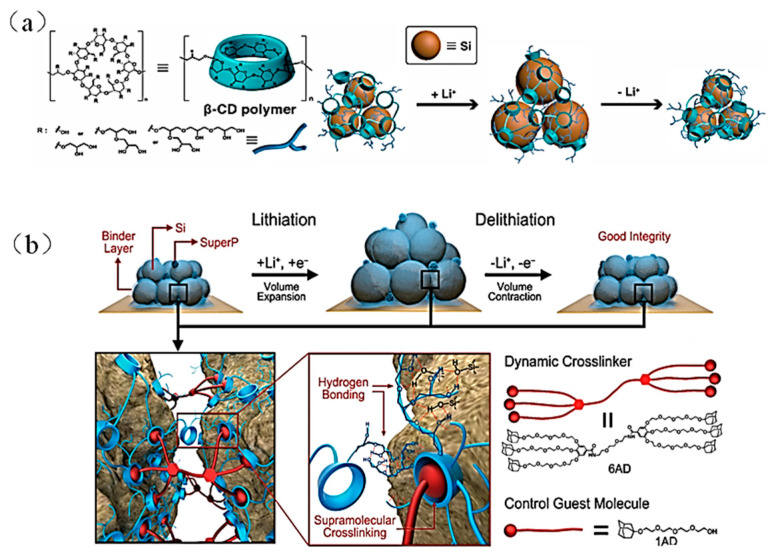
(**a**) Hyperbranched network structure of β-CDp and the mechanism in the process of continuous volume change in the Si anode [90]. (**b**) Dynamic cross-linking mechanism of β-CDp and 6AD during volume change in Si anode and the molecular structure of 6AD [66].

**Figure 20 materials-16-04266-f020:**
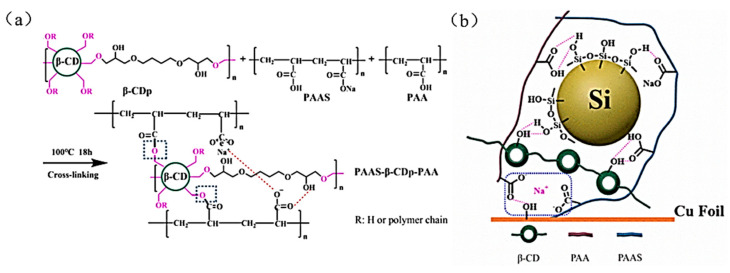
(**a**) Reaction mechanism of PAAS-β-CDp-PAA binder. (**b**) Bonding of PAAS-β-CDp-PAA binder to Si and the Cu collector [39].

**Figure 21 materials-16-04266-f021:**
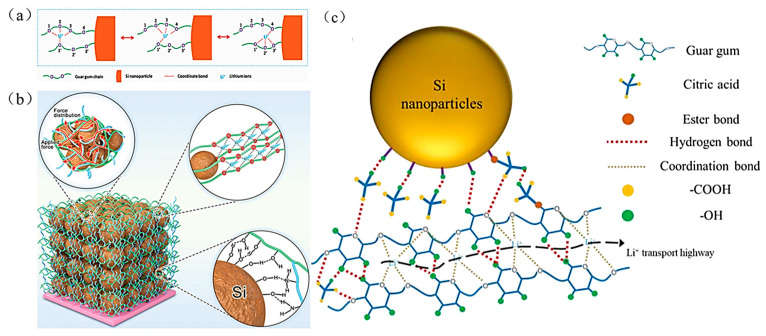
(**a**) Schematic diagram of lithium-ion migration in GG binder [91]. (**b**) Mechanism of GG-g-PAM binder in Si electrode [45]. (**c**) Schematic diagram of chemical interaction between GCA13 binder and Si [36].

**Figure 22 materials-16-04266-f022:**
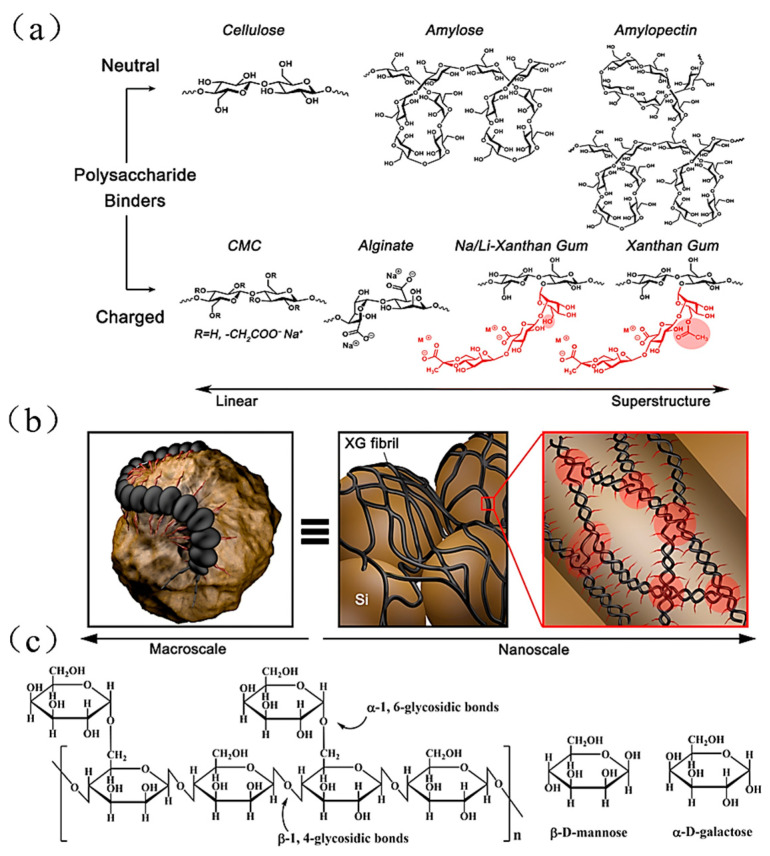
(**a**) Classification of polysaccharide polymers based on electrostatic charge and molecular structure. (**b**) Macroscopic-to-microscopic XG binder adhesion effect on Si particles [92]. (**c**) Formula for the molecular structure of sesbania gum [37].

**Figure 23 materials-16-04266-f023:**
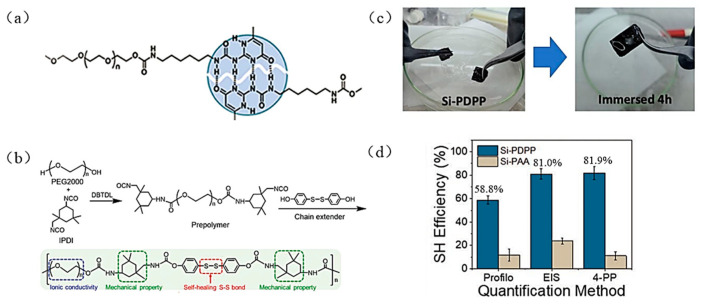
(**a**) Self-healing mechanism of the Upy-PEG-Upy binder [67]. (**b**) Synthesis of BFPU polymer and the specific role of different molecular structures [68]. (**c**) Self-healing test of Si-PDPP anode in the electrolyte (**d**) Three methods quantify the self-healing efficiency of the Si anode with binder [38].

**Figure 24 materials-16-04266-f024:**
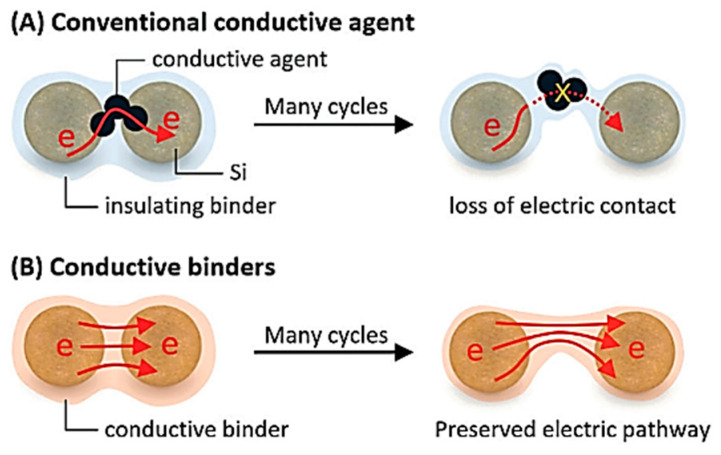
(**A**) Working principle of conventional Si anode using conductive agent. (**B**) Working principle of Si anode using conductive binder [6].

**Figure 25 materials-16-04266-f025:**
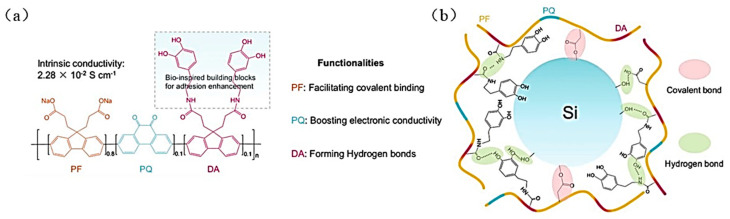
(**a**) Molecular structure of PFPQDA and functional description of each part in the monomer unit. (**b**) Mechanism of PFPQDA binder with Si particles [35].

**Table 1 materials-16-04266-t001:** Suggestions for enhancing Si anode performance.

Suggestion	Specific Implement Scheme	Literature
1. Modification and compounding of active materials.	Active-inactive alloy system, Silicon nanoparticles, Silicon nanowires, Silicon nanotubes, Si/C composites, SiO_x_ materials, etc.	[12,20,22,23,24,25,26,27,28,29]
2. The use of electrolyte additives.	Fluoroethylene carbonate (FEC), Phosphorous-containing compounds, etc.	[17,18]
3. The use of binder for Si anode.	Polyvinylidene fluoride (PVDF), Poly (acrylic acid) (PAA), Sodium carboxymethylcellulose (CMC), Alginate (Alg), Chitosan (CS), β-cyclodextrin polymer (β-CDp), Guar gum (GG), Self-healing polymer, Conducting polymers, etc.	[30,31,32,33,34,35,36,37,38,39]

**Table 2 materials-16-04266-t002:** A summary of properties of different Si-based anode polymer binders.

Type of Binder	Active Material	Solvent	Binder: Conductive Agent: Active Material	Initial Coulomb Efficiency	Current Density (mA g^−1^)	Cycling Performance	Capacity Retention	Ref.
PVDF-b-PTFE	Si powder (1–5 um)	NMP	5:15:80	82.70%	4200	~1000 mAh g^−1^ after 250 cycles	-	[51]
Si-PAA-C	Si nanoparticles	Tris buffer solution	20:20:60	71.20%	2100	~1700 mAh g^−1^ after 400 cycles	48.57%	[52]
PAA-PVA	Si nanoparticles (30–100 nm)	Water	20:20:60	83.90%	400	2283 mAh g^−1^ after 100 cycles	63.14%	[53]
PAA-PEGPBI	Si nanoparticles /graphite (3:7)	NMP	10:10:80	87.30%	130	751.0 mAh g^−1^ after 100 cycles	58.13%	[54]
PR-PAA	Si micron particles (~2 um)	DMSO	10:10:80	91.22%	600	2.43 mAh cm^−2^ after 150 cycles	91%	[55]
NaPAA-g-CMC	Si nanoparticles (50–100 nm)	Water	20:20:60	84%	840	1816 mAh g^−1^ after 100 cycles	79.30%	[56]
PAA-VTEO	Si/graphite	Water	3:7:90	89.40%	47	466 mAh g^−1^ after 100 cycles	99.19%	[57]
PAA-SS	Si nanoparticles	Water	20:20:60	93.20%	4200	1559 mAh g^−1^ after 150 cycles	-	[33]
PE55	Si nanoparticles (30–50 nm)	Water	20:20:60	89.70%	1000	2322.2 mAh g^−1^ after 100 cycles	-	[58]
c-CMC-CA	Core–Shell Carbon/Si	Water	20:20:60	74%	1000	1640 mAh g^−1^ after 100 cycles	87.70%	[59]
CMC-PEG	Silicon powder (≤50 nm)	Water	10:5:85	81%	1786	~2000 mAh g^−1^ after 350 cycles	-	[60]
CMC/EDTA-Ca^2+^	Si/graphite	Water	15:7.5:77.5	81.10%	500	776 mAh g^−1^ after 200 cycles	87.00%	[31]
LiCMC-TA	Si nanoparticles (~100 nm)	Water	1:9:90	80.65%	1000	1701 mAh g^−1^ after 150 cycles	80.00%	[61]
Ca-Alg	Si/C composite	Water	29:18:53	-	420	1822 mAh g^−1^ after 120 cycles	82.30%	[62]
Al/Ba-alg	Si nanoparticles	Water	20:20:60	-	420	~2100 mAh g^−1^ after 300 cycles	-	[63]
PAA-SA	Si-Al alloys (Si:Al = 8:92)	Water	22.5:7.5:70	80.10%	1000	~1419.8 mAh g^−1^ after 200 cycles	48.46%	[30]
CS/ENR	Si nanoparticles (~100 nm)	Water	15:15:70	81.40%	8000	1350 mAh g^−1^ after 1600 cycles	-	[64]
CS-EDTA	SiO_x_ (Si:SiO_2_ = 1:2)	Water	10:20:70	62.80%	1000	721 mAh g^−1^ after 200 cycles	78%	[65]
β-CDp/6AD	Si nanoparticles (~50 nm)	Water	20:20:60	84%	1500	~1500 mAh g^−1^ after 150 cycles	90%	[66]
PAAS-β-CDp-PAA	Si nanoparticles	Water	20:20:60	89.79%	200	2513 mAh g^−1^ After 100 cycles	71.10%	[39]
GG-g-PAM	Si nanoparticles	Water	10:10:80	87.50%	1000	1687 mAh g^−1^ after 200 cycles	71.80%	[45]
GCA13	Si nanoparticles (~100 nm)	Water	15:5:80	93%	2000	1184 mAh g^−1^ after 740 cycles	-	[36]
SG	Si nanoparticles (80–100 nm)	Water	20:20:60	90.92%	1000	2023 mAh g^−1^ after 120 cycles	-	[37]
UPy-PEG-UPy	Si nanoparticles (50–70 nm)	Tetrachloroethane	15:25:60	81%	-	1454 mAh g^−1^ after 400 cycles	84%	[67]
PAA-BFPU	SiOx material	Water	15:15:70	~89%	1200	~3000 mAh g^−1^ after 100 cycles	97%	[68]
PDPP	Si powder (<100 nm)	Water	15:10:75	~94%	2415	2312 mAh g^−1^ after 100 cycles	84%	[38]
PAA–PANI	Si nanoparticles (~100 nm)	Water	25:75 (no conductive agent)	-	4200	~1118 mAh g^−1^ after 300 cycles	56.50%	[69]
PEDOT:PSS	Si nanoparticles (50–70 nm)	Water	10:10:80	~78%	1000	1950 mAh g^−1^ after 100 cycles	~75%	[70]
CG	Si nanoparticles	Water	10:90 (no conductive agent)	80%	840	1500 mAh g^−1^ after 700 cycles	-	[71]
PPTU	Si nanoparticles	Water	10:80 (no conductive agent)	80.10%	1000	2081 mAh g^−1^ after 300 cycles	~76.7%	[72]
PFPQDA	Si nanoparticles (~50 nm)	Water	10:20 (no conductive agent)	72.30%	420	2523 mAh g^−1^ after 150 cycles	96%	[35]

## Data Availability

Data are available in the source publications listed in the bibliography.

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
