# Peer review of "Application and Development of Silicon Anode Binders for Lithium-Ion Batteries"

_materials, 2023, doi:10.3390/ma16124266_

Round 1

Reviewer 1 Report (Previous Reviewer 1)

The authors properly answered the questions. The manuscript can be published in Materials.

Author Response

Thank you again for your suggestions. All your suggestions have important guiding significance for my article writing and scientific research work!

Reviewer 2 Report (Previous Reviewer 2)

Authors have significantly improved the manuscript in this revised form

Author Response

Thank you again for your suggestions. All your suggestions have important guiding significance for my article writing and scientific research work!

Reviewer 3 Report (Previous Reviewer 3)

A number of references should be cited (authors must check the literature and cite the relevant refs) to improve section 3: suggestions for enhancing Si anode performance.

1. In section 3: Suggestions for enhancing Si anode performance: Authors summarized several suggestions on how to overcome issues with Si material by employing nanostructure, C/Si and other composite systems. However, I am still not convinced by the way they summarized these improvements. Especially, there has been a lot of investigations on the active/inactive amorphous-Si/FeSi2 and graphite composite. With the inclusion of iron disilicide (FeSi2), the volume expansion of the Si can be controlled resulting in higher structural stability and long term cycling stability. Authors must check the other references together with these references to cite in the suggestions for enhancing Si anode performance.

1.     Vorauer, T., Kumar, P., Berhaut, C.L. et al. Multi-scale quantification and modeling of aged nanostructured silicon-based composite anodes. Commun Chem 3, 141 (2020). https://doi.org/10.1038/s42004-020-00386-x

2.     Kumar, P., Berhaut, C. L., Zapata, D., De, E., Tardif, S., Pouget, S., Lyonnard, S., Jouneau, P.-H., Nano-Architectured Composite Anode Enabling Long-Term Cycling Stability for High-Capacity Lithium-Ion Batteries. Small 2020, 16, 1906812. https://doi.org/10.1002/smll.201906812

2.     The authors nicely summarizes the various Si anode and binder systems for stable Li-ion batteries. Even they mentioned see page 15 “In 2019, Lee et al. copolymerized CMC with polyethylene glycol (PEG) to obtain the CMC-PEG binder [57] (Figure 13a), which is rich in carboxyl, ether and hydroxyl groups that can offer some bonding sites in three dimensions form hydrogen and ester bonds with the active material Si surface to effectively improve the bonding between electrode materials (Figure 13b).The Si anode made with this binder is quite stable which displays a specific capacity of almost 2000 mAh g-1 after 350 cycles”.

In my opinion, 350 cycles for LiBs are not sufficient. Therfore, I recommened the authors to check the relavent litertaure and add a few applications of the Si/anode binder system for long-term or extended cycles. Long term mechanical and structural stabilty is one of the main challenges with Si based anodes materails.

3.     Another important aspect for Si anodes is aging mechanisms, I did not see much information in the article that eludicate the aging mechaisms in Si anodes. It is advisable to add a few examples by adding morphological, structural and chemical changes that appear after the aging and how it impacts the overall performace of the Si anodes.

Author Response

Dear reviewer,

We feel great thanks for your professional review work on our article entitled “Application and development of silicon anode binder for lithium-ion batteries” (ID: materials-2413380). Those comments are valuable for revising and improving our manuscript. We have read through comments carefully and have made extensive corrections which we hope meet with approval. The main corrections in the manuscript and the responds to the reviewer's comments are as flowing. The reviewer comments are laid out below in italicized font and specific concerns have been numbered. The serial number is highlighted in green. We sincerely appreciate the valuable comments about supplement of literatures. We have checked the literature carefully and added references on enhancing Si anode performance in the revised manuscript.

Responds to the reviewer's comments:

  1. Response to comment: “In section 3: Suggestions for enhancing Si anode performance…”

Response: We are grateful for the suggestion. In section three suggestions for enhancing the of Si anode performance, the method of using active/inactive alloy system materials to improve the performance of silicon anode is added. In addition, the active/inactive alloy system components and properties are introduced in detail.

  1. Response to comment: “In my opinion, 350 cycles for LiBs are not sufficient…”

Response: We sincerely thank the reviewer for careful reading and suggestion.    In order to better explain that the use of polymer binder can improve the cycling stability of Si anode. It is mentioned in the fourth part of the specific introduction of binder that the use of polymer binder can make the cycle of Si anode more than 1000 times with well performance.

  1. Response to comment: “Another important aspect for Si anodes is aging mechanisms…”

Response: We appreciate the suggestion. In order to make the article more complete, we added the specific aging mechanism of the Si anode and clear SEM images. It can be clearly seen from the microscopic image that the continuous expansion and contraction of the silicon volume in the electrode cycle process leads to cracks in the silicon particles and finally damages the electrode structure.

We tired our best to improve the manuscript and any changes we make to the manuscript use the “Track Changes” function. We appreciate for reviewer’s warm work earnestly, and hope the correction will meet with approval. Once again, thank you very much your comments and suggestions.

Sincerely,

Zheng Chen and Danming Chao

Jilin University

Xiuzheng Road 1788, Changchun 130012, China

E-mail: [email protected], [email protected]

Reviewer 4 Report (New Reviewer)

The authors reviewed various types of reactive and non-reactive anode binders for Si anode. The manuscript includes a short back ground on Si anode, pros and cons of Si, suggestions to improve the performance of Si anode, anode binders for Si, and future outlook. Since Si is considered as a potential high capacity anode for the future high energy LIBs, the detailed review on anode binder for Si will be useful to scientific community. Moreover, the manuscript is organized carefully. I would recommend for publication after minor revision.

1. Si-based binder implies that the binder contains Si. The authors may correct it as " Binder for Si anode"

Authors are requested to make thorough check for typos and grammatical errors

Author Response

Dear reviewer,

We feel great thanks for your professional review work on our article entitled “Application and development of silicon anode binder for lithium-ion batteries” (ID: materials-2413380). Those comments are valuable for revising and improving our manuscript. We have read through comments carefully and have made corrections which we hope meet with approval. The main corrections in the manuscript and the responds to the reviewer's comments are as flowing. The reviewer comments are laid out below in italicized font and specific concerns have been numbered. The serial number is highlighted in green.

Responds to the reviewer's comments:

  1. Response to comment: Si-based binder implies that the binder contains Si. The authors may correct it as "Binder for Si anode".

Response: We are grateful for the suggestion. We apologize for not expressing ourselves clearly. The language presentation was improved with assistance from a professional with appropriate research background. We have revised the corresponding part of the manuscript on ”Si-based binder “changes “binder for Si anode” have been made to make the expression clearer and more accurate.

We tired our best to improve the manuscript and any changes we make to the manuscript use the “Track Changes” function. We appreciate for reviewer’s warm work earnestly, and hope the correction will meet with approval. Once again, thank you very much your comments and suggestions.

Sincerely,

Zheng Chen and Danming Chao

Jilin University

Xiuzheng Road 1788, Changchun 130012, China

E-mail: [email protected], [email protected]

Reviewer 5 Report (New Reviewer)

Shen et.al., reviewed different binders for silicon as Li-ion battery anode. Details of the binding mechanism of different binders are discussed in detail. The manuscript is of interest but there are important aspects missing which is not providing sufficient interest for battery community. The manuscript may be considered after a major revision.

1. The detailed binding mechanism is of utmost important however, without correlating the battery performance for each case it makes it more chemistry.

2. Review of experimental data each case especially comparison of different binders for varying size from nano to bulk silicon would be useful.

3. There are recent reviews on binder for silicon (example: 10.1007/s12274-022-5281-7 and there are more) with mechanism and correlation with performance in such a scenario what is the importance differences between the published and this review should be described properly.

4. In several cases Coulombic efficiency (CE) is noted only in the first cycle however, in subsequent cycles also should be given importance.  So CE as parameter for longer cycling should be an important criterion for indicating the quality of binders.

5. Solvent for binders is important, battery community is interested in water as solvent for binders more such binders may be highlighted and described with comparison with organic solvent binders.

Minor corrections only required.

Author Response

Dear reviewer,

We feel great thanks for your professional review work on our article entitled “Application and development of silicon anode binder for lithium-ion batteries” (ID: materials-2413380). Those comments are valuable for revising and improving our manuscript. We have read through comments carefully and have made extensive corrections which we hope meet with approval. The main corrections in the manuscript and the responds to the reviewer's comments are as flowing. The reviewer comments are laid out below in italicized font and specific concerns have been numbered. The serial number is highlighted in green. We sincerely appreciate the valuable comments about supplement of literatures. We have checked the literature carefully and added references on enhancing Si anode performance in the revised manuscript.

Responds to the reviewer's comments:

  1. Response to comment: “The detailed binding mechanism…”

Response: We are grateful for the suggestion. The different bonding mechanism of binders acting on the silicon anode makes the anode show various properties. In order to clearly express the different improvement of battery performance through different bonding mechanisms with different binders, we modified the Table 2 to give specific battery performance.

  1. Response to comment: “Review of experimental data each case especially comparison of different binders for varying size from nano to bulk silicon would be useful.”

Response: We sincerely thank the reviewer for careful reading and suggestion. During battery charging and discharging, the volume of silicon will expand and contract that cause the silicon particles to crack. The amount of silicon cracking will be somewhat reduced when the size of the active material silicon is down to the nanoscale size. Therefore, different sizes of silicon in the Si-based anode will show different results. In order to better display the data, we have supplemented the size of the active substance in Table 2.

  1. Response to comment: “There are recent reviews on binder for silicon…”

Response: We appreciate the suggestion. In this manuscript the polymer binder is separated into two primary types of binders that rely on physical contact (intermolecular forces) and chemical interaction (hydrogen and chemical bond). Our manuscript differs from other articles in the classification of the mechanism and properties of the binder. The specific differences are briefly described in section four.

  1. Response to comment: “In several cases Coulombic efficiency (CE) is noted only in the first cycle however…”

Response: We are grateful for the suggestion. Coulomb efficiency is necessary for battery performance. Not only should the first coulomb efficiency be given when we describe the battery performance, but also the coulomb efficiency after different cycles of the battery. Therefore, we read different literature again and carefully read the literature description of coulomb efficiency in battery performance and added it to the manuscript.

  1. Response to comment: “Solvent for binders is important, battery community is interested in water as solvent…”

Response: We sincerely thank the reviewer for careful reading. At present, organic solvents such as NMP are commonly used as solvents in batteries. But in the actual production application, water as a solvent for binder is environmentally friendly. In order to describe the solvent type of binder more clearly, solvents of different binders are added in Table 2.

We tired our best to improve the manuscript and any changes we make to the manuscript use the “Track Changes” function. We appreciate for reviewer’s warm work earnestly, and hope the correction will meet with approval. Once again, thank you very much your comments and suggestions.

Sincerely,

Zheng Chen and Danming Chao

Jilin University

Xiuzheng Road 1788, Changchun 130012, China

E-mail: [email protected], [email protected]

Round 2

Reviewer 5 Report (New Reviewer)

The revised review may be accepted for publication.

This manuscript is a resubmission of an earlier submission. The following is a list of the peer review reports and author responses from that submission.

Round 1

Reviewer 1 Report

In this review “Application and development of silicon anode binder for lithium-ion batteries” the authors report the research work on the design and development of new Si-based anode binders able to improve the cycling stability of the anode structure, summarizing and outlining the progress of this research topic. The topic is interesting, however some recent works focus on innovative silicon anode binders, such as: doi.org/10.1016/j.jpowsour.2020.229331, doi.org/10.1016/j.cej.2020.126807, doi.org/10.1002/aenm.202200850.

In general both the scientific and English languages are very poor and lacking precision. In particular the introduction part should be strongly reviewed. Unfortunately, this work is quite incomplete, the quality of the figures is poor and last but not least the multiple English mistakes and grammatical errors make the whole paper quite difficult to understand. For these reasons, in my opinion, this work is not suitable for publication in Materials.

Below some examples and comments:

Page 2: “Following is the procedure…”

“…so that they can easily escape from the electrode and spread into the electrolyte as well as destabilize the electrode structure”

“…been demonstrated to guard against electrolyte infiltration”

Etc…

In addition, Table 1 should be reviewed, because it seems confused and not clear.

The quality of the figures should be improved, some details are not visible.

Page 11: “with a capacity retention rate” – the capacity retention is not a rate.

Some typo issues should be corrected.

A table comparing the different cited binders, useful parameters, and electrochemical results should be provided in order to make more clear the discussion.

Reviewer 2 Report

This review deals with recent advances on various binders for Si-based anodes according to their structures, physical and chemical bonding, self-healing capabilities and so. The subject matter is of general interest. However, such topic has been extensively developed in the literature and this work is not among the most interesting and well-structured reviews that have been published in recent years.

Down below, I report some reviews on the subject of binders for the silicon anode and the Author can cite the areas by not mentioning any specific reference. The only references that should be cited are: “The progress of novel binder as a non-ignorable part to improve the performance of Si-based anodes for Li-ion batteries “https://doi.org/10.1002/er.3826” and “Critical roles of binders and formulation at multiscales of silicon-based composite electrodes” https://doi.org/10.1016/j.jpowsour.2015.01.140, that are consistent to this review article.

In general, this review is quite well structured, but it needs extensive editing of English language and style, especially in the first part (from page 2 to 8).  There are many mistakes and typos, which make reading this very difficult.

All the figure captions must report the reference number from which they were taken.  

In particular, the parts that need to be reviewed carefully are:

Page 2: use “complex reactions” instead of “intricate reactions”

Page 3 : table 1 – FEC is Fluoroethylene carbonate

Page 4: use “capacity fading” instead of “reversible capacity declines”

Page 4: revise this sentence: “Researchers added electrolyte additives FEC to EC and DEC solvents,

which can react preferentially to form an SEI layer than other electrolytes due to their

higher reduction potential of decomposition. The stable SEI layer limits the appearance of

large cracks on the Si surface, while inhibiting the generation of additional SEI layer by

other solvents” (Which are the other solvents?)

Page 4: “Therefore, one of the crucial strategies to increase the cycle stability of Si anode is to choose to use electrolyte additives”  (to choose or to use? please, choose a verb)

Page 4: revise the sentence in English and with more correct terms  : “According to Figure 3, the active material, conductive agent, and polymer binder make up the battery electrodes. The active material is used to store lithium ions, which is the primary determinant of battery capacity; the use of a conductive agent improves the conductivity of the electrode; The crucial function of the polymer binder is to tightly bond the active material, a conductive agent, and current collector to keep the stability of the

electrode structure during the charging and discharging process”

Page 5: “The Si-based binder needs to play an excellent role in Si anode ….” Revise the sentence

Page 5: “The following Si-based anode polymer binder is introduced in this study “-Revise the sentence

Figure 4 : in the center of the figure please revise: “polymer binders”

Page 5: "the desolution/diffusion/penetration" step: “the dissolution/diffusion/penetration”?

Page 6-7: these parts need extensive English editing; sentences are too long and difficult to understand

Page 8: “….the battery to better sustain the Si anode structure’s the stability during insertion/deinsertion processes” this sentence must be revised.

Page 8: “Inspired by the research work on dopamine in mussel foot proteins, to enhance the interaction between PAA binder and Si surface, in 2013….” Please revise the sentence.

Page 9 “When the Si anode in charge-discharge cycles, the α-CD slides along the polyethylene glycol (PEG) backbone” What does it mean?

Please use:  “ capacity retention” instead of “capacity retention rate” through the whole manuscript.

Na-CMC is CMC, please use always the same acronym

Pag 14 “which improved electrochemical performance while being able to endure the significant state change of the silicon anode” What is the “significant state change”?

Pag 14 What does it means “after 150 cycles density”?

Pag 14 please revise the sentence “Furthermore, natural binders are plentiful, inexpensive, environmentally friendly, and environmentally friendly, making them become potential Si anode binders”. Environmentally friendly appears twice.

Pag 16: please revise the sentence: “at an active substance Si loading ..”

Revise the caption of figure 19

5. Conclusion and outlook

All titles should be written in bold and subtitles in italics

 Here some of the recent review on this topic, that were not cited in this work:

Recent Progress on Polymeric Binders for Silicon Anodes in Lithium-Ion Batteries Journal of Electrochemical Science and Technology 2015;6(2):35-49.

DOI: https://doi.org/10.5229/JECST.2015.6.2.35

Key Factors for Binders to Enhance the Electrochemical Performance of Silicon Anodes through Molecular Design

https://doi.org/10.1002/smll.202101680

Molecular design principles for polymeric binders in silicon anodes

https://doi.org/10.1039/C9ME00162J

Confronting the Challenges of Next-Generation Silicon Anode-Based Lithium-Ion Batteries: Role of Designer Electrolyte Additives and Polymeric Binders

https://doi.org/10.1002/cssc.201900209

Biomass-derived polymeric binders in silicon anodes for battery energy storage applications

DOI        https://doi.org/10.1039/D1GC01814K

Recent progress and future perspective on practical silicon anode-based lithium ion batteries

https://doi.org/10.1016/j.ensm.2022.01.042

A review of rational design and investigation of binders applied in silicon-based anodes for lithium-ion batteries

https://doi.org/10.1016/j.jpowsour.2020.229331

Challenges and recent progress on silicon‐based anode materials for next‐generation lithium‐ion batteries

https://doi.org/10.1002/sstr.202100009

The progress of novel binder as a non-ignorable part to improve the performance of Si-based anodes for Li-ion batteries

https://doi.org/10.1002/er.3826

Silicon anodes for high‐performance storage devices: structural design, material compounding, advances in electrolytes and binders

https://doi.org/10.1002/cnma.201900708

Critical roles of binders and formulation at multiscales of silicon-based composite electrodes

https://doi.org/10.1016/j.jpowsour.2015.01.140

Polymer Binders Constructed through Dynamic Noncovalent Bonds for High-Capacity Silicon-Based Anodes

https://doi.org/10.1002/chem.201900988

Intermolecular chemistry for designing functional binders in silicon/carbon composite anodes

https://doi.org/10.1016/j.mtener.2022.101153

A review of existing and emerging binders for silicon anodic Li-ion batteries

https://doi.org/10.1007/s12274-022-5281-7

Reviewer 3 Report

The review article "Application and development of silicon anode binders for lithium-ion batteries" focuses on the development of new Si-based anode binders to improve cycling stability. As Si has many challenges, starting from its volume expansion to the unstable SEI formation, and finally the mechanical fracture and rapid capacity fading. There is a dire need to search for alternative binder materials that can provide better mechanical stability during the charge/discharge cycle. The author’s efforts in compiling and providing various binder-based materials for understanding are of interest. However, I feel that a number of good review articles have already been published on this subject. I refer to a review article published in 2021 that nicely summarizes the advances of various binder materials, especially for the silicon anodes. In view of the above arguments, I do not feel that this article contributes to the advancement of science.

I would reject this article in its present form.

Also, the authors have used various figures from different sources or articles, but they did not give proper citation to all these figures, which is ethically not encouraging. Considering the reputation of Materials Journal, I won’t recommend this article for publication.

Advances of polymer binders for silicon-based anodes in high energy density lithium-ion batteries, https://doi.org/10.1002/inf2.12185